# A compressed hierarchy for visual form processing in the tree shrew

Frank F. Lanfranchi[1,2 ✉], Joseph Wekselblatt[2], Daniel A. Wagenaar[2,3] & Doris Y. Tsao[1,4 ✉]

Our knowledge of the brain processes that govern vision is largely derived from studying primates, whose hierarchically organized visual system[1] inspired the architecture of deep neural networks[2]. This raises questions about the universality of such hierarchical structures. Here we examined the large-scale functional organization for vision in one of the closest living relatives to primates, the tree shrew. We performed Neuropixels recordings[3,4] across many cortical and thalamic areas spanning the tree shrew ventral visual system while presenting a large battery of visual stimuli in awake tree shrews. We found that receptive field size, response latency and selectivity for naturalistic textures, compared with spectrally matched noise[5], all increased moving anteriorly along the tree shrew visual pathway, consistent with a primate-like hierarchical organization[6,7]. However, tree shrew area V2 already harboured a high-level representation of complex objects. First, V2 encoded a complete representation of a high-level object space[8]. Second, V2 activity supported the most accurate object decoding and reconstruction among all tree shrew visual areas. In fact, object decoding accuracy from tree shrew V2 was comparable to that in macaque posterior IT and substantially higher than that in macaque V2. Finally, starting in V2, we found strongly face-selective cells resembling those reported in macaque inferotemporal cortex[9]. Overall, these findings show how core computational principles of visual form processing found in primates are conserved, yet hierarchically compressed, in a small but highly visual mammal.

The ability to recognize objects is fundamental to the survival of visual animals. The primate ventral stream has long served as a model for studying how objects are processed in the brain[10,11]. One defining feature of the primate ventral stream is hierarchical organization[12], which is mirrored by deep neural networks (DNNs) trained on object recognition[8,13]. This parallel raises an important question: is hierarchical representation necessary and, if so, can it be found across all highly visual mammalian species? Investigating visual processing across different mammalian species promises to provide a deeper understanding of general principles for object vision.

Over a decade ago, the mouse visual system began to attract strong interest, driven by the wealth of tools available for mouse neural circuit dissection[14,15]. However, the mouse's low visual acuity and limited cortical territory dedicated to vision[16] make it a non-ideal organism for studying hierarchical brain mechanisms underlying object recognition. The tree shrew has attracted growing interest as a model to study visual processing[17] owing to its high visual acuity (more than ten times that of rodents)[18], greatly expanded visual cortex[19] and excellent ability to perform visually guided behavioural tasks compared with the mouse[20,21]. The tree shrew visual system includes at least nine distinct anatomical visual cortical areas[19]. The primary visual area (V1) shows a high degree of functional specialization, including an orderly arrangement of orientation-selective columns[22,23]. The tree shrew also has a prominent

second visual area (V2), albeit with a large-scale topographic organization that differs from that of primates[24]. Lesion studies suggest a rough correspondence between tree shrew extrastriate areas anterior to V2 and primate IT cortex: ablations of large portions of the temporal lobe produce deficits in pattern discrimination and object vision similar to the effects of inferotemporal (IT) lesions in primates[19,25,26]. However, to our knowledge, there have been no electrophysiological studies of the functional properties of tree shrew extrastriate visual areas beyond V2.

Here we aim to identify the cortical organization and coding principles that underlie visual object representation across the entire tree shrew ventral stream. Using large-scale electrophysiological recordings with several Neuropixels probes, we surveyed five tree shrew ventral visual areas as well as the pulvinar. We confirmed hallmarks of hierarchical organization found in primates, including increased receptive field size and response latency[27] as well as increased selectivity for naturalistic textures compared with spectrally matched noise[5]. We found that area V2 in the tree shrew performs key functions associated with primate IT cortex. This includes a full representation of high-level object space, accurate object identity decoding and reconstruction, and the presence of strongly face-selective cells. Overall, the results indicate a compressed, multi-stage hierarchy in the tree shrew in which representations previously observed in the primate are realized at a much earlier stage of visual processing.

[1]Department of Molecular and Cell Biology, University of California Berkeley, Berkeley, CA, USA. [2]Division of Biology and Biological Engineering and Computation and Neural Systems, Caltech, Pasadena, CA, USA. [3]Caltech Neurotechnology Laboratory, Caltech, Pasadena, CA, USA. [4]Howard Hughes Medical Institute, University of California Berkeley, Berkeley, CA, USA. ✉e-mail: flanfran@caltech.edu; dortsao@berkeley.edu

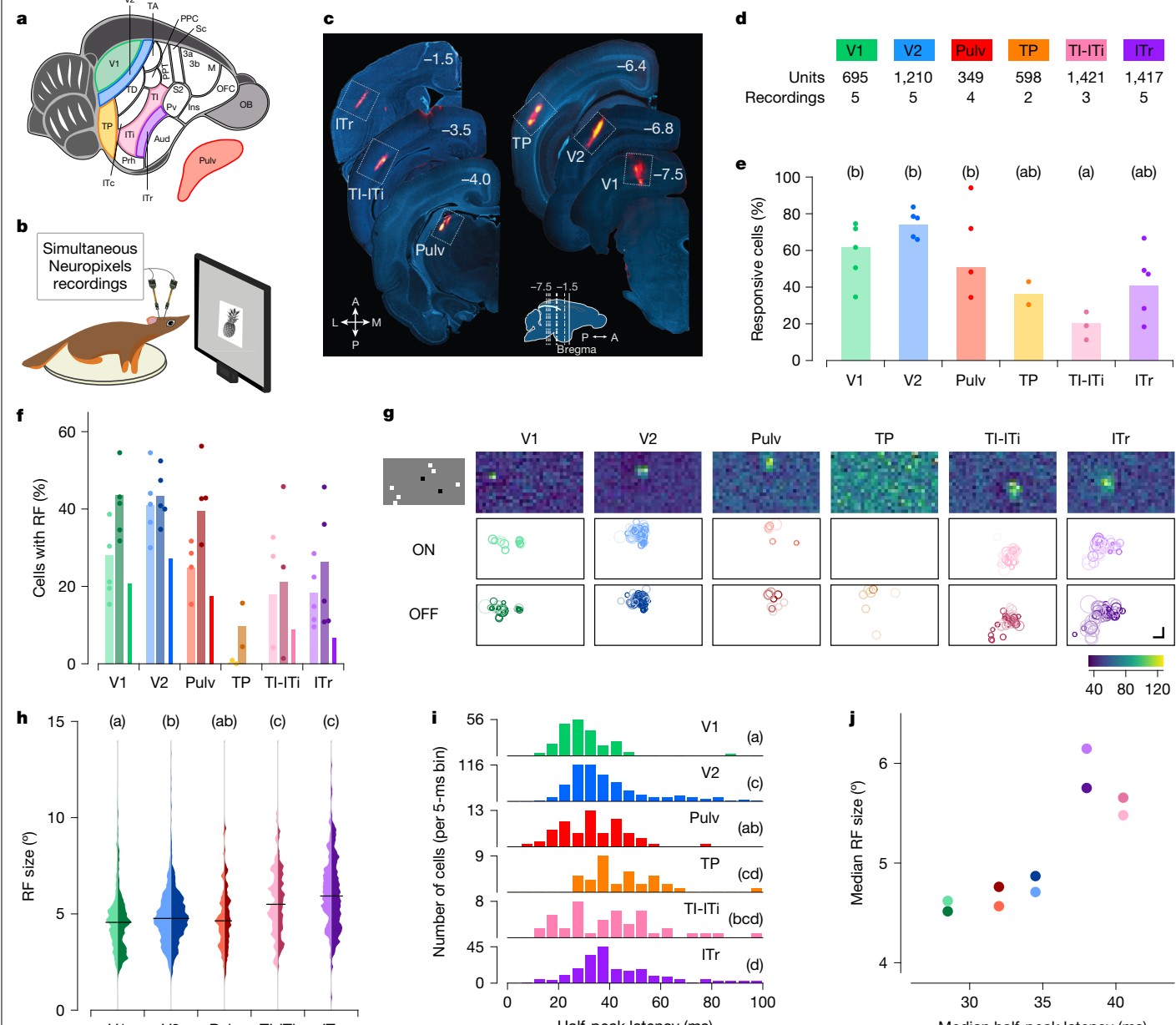

**Fig. 1 | High-throughput electrophysiological recordings along the tree shrew ventral visual pathway reveal a functional hierarchy. a,b**, Schematic of a tree shrew brain (**a**) and head-fixed electrophysiological recordings with Neuropixels probes (**b**). **c**, Representative electrode tracks marked with DiI (red) in each targeted area. Numbers indicate rostrocaudal position relative to Bregma (inset). A, anterior; L, lateral; M, medial; P, posterior. **d**, Number of recordings and total units across each area. **e**, Percentage of visually responsive cells to any of the presented visual stimuli in each area (Methods). Dots indicate individual recordings, bars indicate averages across recordings. Letters in this and subsequent panels indicate Tukey grouping. Tukey analysis ($\alpha = 0.05$) after ANOVA, $F_{5,18} = 5.362$, $P = 0.003$). **f**, Percentage of visually responsive units (see Fig. 1e) showing receptive fields (RFs). Left (lighter, ON), centre (darker, OFF) and right (ON/OFF) bars for each area. Dots indicate individual recording sessions. **g**, Distribution of receptive field locations across the visual field. Top row, receptive field maps for example units, one per area. Middle and bottom rows show the position and sizes of all ON and OFF receptive fields (respectively) in a representative recording. Shading indicates receptive field quality (Methods). Each white box represents ±54° horizontally and ±38° vertically. Top left, one frame of sparse noise stimulus used to map receptive fields. **h**, Distribution of ON (left, lighter) and OFF (right, darker) receptive field sizes for each area. Tukey analysis ($\alpha = 0.05$) after ANOVA, $F_{4,1540} = 36.544$, $P < 10^{-28}$; TP was excluded from this analysis because of the very low number of cells with receptive fields in this area. **i**, Histogram of the latencies to half-peak response in visually responsive cells in each area. Tukey analysis ($\alpha = 0.05$) after ANOVA $F_{5,1147} = 20.197$, $P < 10^{-18}$. **j**, Comparison of the hierarchy inferred from receptive field size (*y* axis) and response latency (*x* axis). Each dot represents the median of the data for a given area (hue), with ON and OFF receptive fields represented by light and dark dots, respectively. Scale bar, 15°.

We targeted a set of areas spanning the tree shrew ventral stream to investigate hierarchical visual processing (Fig. 1a). We included primary (V1) and secondary (V2) visual areas as architectonically distinct regions involved in early stages of visual processing[28,29]. As a potential intermediate node along the ventral visual processing stream, we selected the temporal posterior (TP) area. At the anterior end, we focused on three subregions that may be homologous to macaque IT cortex: temporal-inferior (TI), temporal intermediate (ITi) and inferotemporal rostral (ITr) areas. Lesions to TI and ITi cause drastic impairments in visual form detection[25]. ITr receives inputs from both visual and auditory cortex[19], but its visual functional properties have never been explored, to our knowledge. Owing to the difficulty in distinguishing

the border between TI and ITi, we grouped them together and refer to this region as TI-ITi. Because many temporal areas receive direct input from the thalamus[30], we also included the dorsal visual portion of the pulvinar (Pulv) in our recordings. To guide electrode targeting, we performed retrograde tracing experiments (Extended Data Fig. 1a,b).

To characterize the visual responses of neurons across V1, V2, TP, TI-ITi, ITr and Pulv, we performed electrophysiological recordings using Neuropixels probes in awake tree shrews (Fig. 1b). During each experiment, animals were head-fixed in front of a monitor and presented with a battery of visual stimuli, including local sparse noise, static gratings, naturalistic textures and noise, and images of faces and other objects. At the conclusion of each session, probe locations were marked with DiI (DiIC18(3), a fluorescent dye) and targeting was confirmed with histology (Fig. 1c). We classified a cell as visually responsive if it responded to any of the classes of visual stimuli we tested (Methods). We found many well-isolated single units in each area (Fig. 1d), with some inter-area differences in the fractions of cells that responded to visual stimuli (ANOVA, $F_{5,18} = 5.362$, $P = 0.003$; Fig. 1e). In particular, significantly fewer TI-ITi cells were visually responsive compared with V2 cells.

We began by mapping the receptive fields of neurons along the tree shrew ventral pathway using a locally sparse noise stimulus (Methods). For each neuron, we estimated the receptive field by fitting a Gaussian distribution to the two-dimensional (2D) matrix of spike counts across visual field locations; ON and OFF receptive fields were computed separately using responses to white and black squares, respectively. Cells with ON and/or OFF receptive fields were clearly present in all areas except TP (Fig. 1f). This included the two most anterior areas TI-ITi and ITr; this contrasts with the anterior temporal lobe in primates, where neurons typically show spatially invariant responses[31,32].

Within individual recordings, receptive field positions were clustered in a small portion of the visual field, corresponding to the retinotopic region represented by the cortical site targeted with the electrode. Figure 1g shows receptive fields of all recorded cells in a representative session for each area. This clustering was evident across all areas studied, including the most anterior areas, TI-ITi and ITr. This finding suggests that, despite their position at the anterior end of the ventral stream, these areas preserve retinotopic organization.

To assess the hierarchical relationships between the recorded areas, we first examined two classic metrics of hierarchical level: receptive field size and visually evoked response latency. Receptive field sizes increased systematically from posterior to anterior (Fig. 1h). We also calculated the half-peak latencies for each unit in each area and found that latencies increased from V1 to V2 to ITr (Fig. 1i and Methods). The hierarchy predicted by the increase in receptive field sizes was broadly consistent with the hierarchy predicted by the increase in latencies (Fig. 1j).

In the primate visual cortex, early visual areas are strongly tuned to low-level features such as orientation and spatial frequency, whereas later areas are tuned to more complex object features[7,33–35]. To examine whether a similar progression exists in the tree shrew, we assessed tuning to orientation and spatial frequency across ventral visual areas using static gratings (Fig. 2a). We found that the proportion of visually responsive neurons (see Fig. 1e) that responded to gratings was the highest in V1 and V2 (roughly 55% and 65%, respectively) and lowest in TI-ITi (Fig. 2b). Tuning to orientation, spatial frequency and phase of example cells from V2 and ITr illustrates the diverse tuning we observed to these variables across tree shrew visual areas (Fig. 2c). Overall, orientation tuning was most prevalent in V1 and V2 (Tukey analysis after ANOVA, $F_{5,1106} = 26.791$, $P < 10^{-24}$, Fig. 2d), whereas spatial frequency tuning was also prevalent in ITr (Tukey analysis ANOVA, $F_{5,1106} = 20.514$, $P < 10^{-18}$, Fig. 2e). These findings are roughly consistent with those found in the primate and rodent ventral stream, where orientation tuning is especially prominent in early visual areas and then sharply decreases in later areas[36–39].

Thus far, V2 responses seemed largely similar to those in V1, raising the question whether V2 performs any distinct computational function. In macaques, sensitivity to higher-order statistical dependencies in naturalistic textures has been identified as a distinguishing feature of area V2 (ref. 5). We therefore asked whether tree shrew extrastriate areas show a similar specialization for naturalistic texture processing. To test this, we recorded neural activity across all six visual areas while presenting naturalistic textures and spectrally matched synthetic noise images (Fig. 2f and Methods). Among all areas, V2 contained the highest proportion of cells that responded to the texture and/or noise stimuli (Fig. 2g). Population response dynamics revealed the strongest differential activity between naturalistic textures and noise in V2, followed by V1, ITr and TI-ITi, with minimal or no modulation in the remaining areas (Fig. 2h). In V2, the difference persisted for the duration of the stimulus. Although responses in V1 commenced well before those in V2 (Fig. 1i), the divergence between texture and noise responses occurred later in V1 (at 90 ms) than in V2 (at 45 ms), suggesting that the texture modulation in V1 may arise through feedback from V2. This interpretation is further supported by the finding that V2 encoded texture family identity earlier than V1 (Fig. 2i).

A central function of the visual hierarchy is to recognize and categorize objects to guide vital behaviours such as navigation, foraging or mating. To investigate high-level object representations in the tree shrew ventral stream, we presented a rich stimulus set consisting of 1,593 images of animals, body parts, faces and everyday objects (Methods). This same stimulus set has previously been used to characterize tuning in macaque inferotemporal (IT) cortex, enabling direct comparisons between object recognition mechanisms in primates and tree shrews[8]. Stimuli were adjusted to match the receptive field location of recorded neurons (Methods). Response rasters from example cells showed diversity in object selectivity across different neurons in the tree shrew ventral stream (Fig. 3a). Among the six areas recorded, a similar proportion of visually responsive cells responded to object stimuli across V2, TP, TI-ITi and Pulv (Fig. 3b). Notably, a much larger fraction of visually responsive cells in TI-ITi responded to object stimuli compared with gratings (Fig. 2b), consistent with temporal areas occupying a higher level in the visual hierarchy. To quantify the reliability of object-driven responses, we computed the 'explainable variance'—the portion of neural response variance attributable to stimulus identity rather than trial-to-trial variability (Methods). After V2, the explainable variance in responses to these complex object stimuli decreased notably (Fig. 3c), indicating that responses in more anterior areas were less consistent across trials. To determine whether the explainable variance could be accounted for by low-level visual features, we analysed the contributions of luminance, contrast and spatial frequency; in each area, only a small fraction of the variance could be explained by such features (Fig. 3c and Extended Data Fig. 2).

To better understand the nature of the neural code used by each area, we modelled neural responses using AlexNet[40], an eight-layer DNN trained on object recognition (Fig. 3d). In macaques, single IT neurons are well described by an 'axis model', in which each cell linearly projects incoming stimuli onto a preferred axis in a DNN-derived feature space[8,13]. In these models, the preferred axes span a relatively low-dimensional basis—such that, for example, just 50 dimensions are sufficient for accurate reconstructions of faces from macaque face patches[41]. To test whether this principle also applies in the tree shrew, we computed the preferred axis of each neuron across six recorded areas using the first 50 principal components from AlexNet layer FC6. We focused on FC6 to clarify whether tree shrew cortex represents a high-level object space, as observed in macaque IT cortex[8]. Consistent with axis-based coding, neurons in all six areas showed ramp-shaped tuning along their preferred axes (Fig. 3e and Methods). Moreover, cells showed flat tuning along their principal orthogonal axis (that is, longest axis orthogonal to the preferred axis; Fig. 3f and Methods).

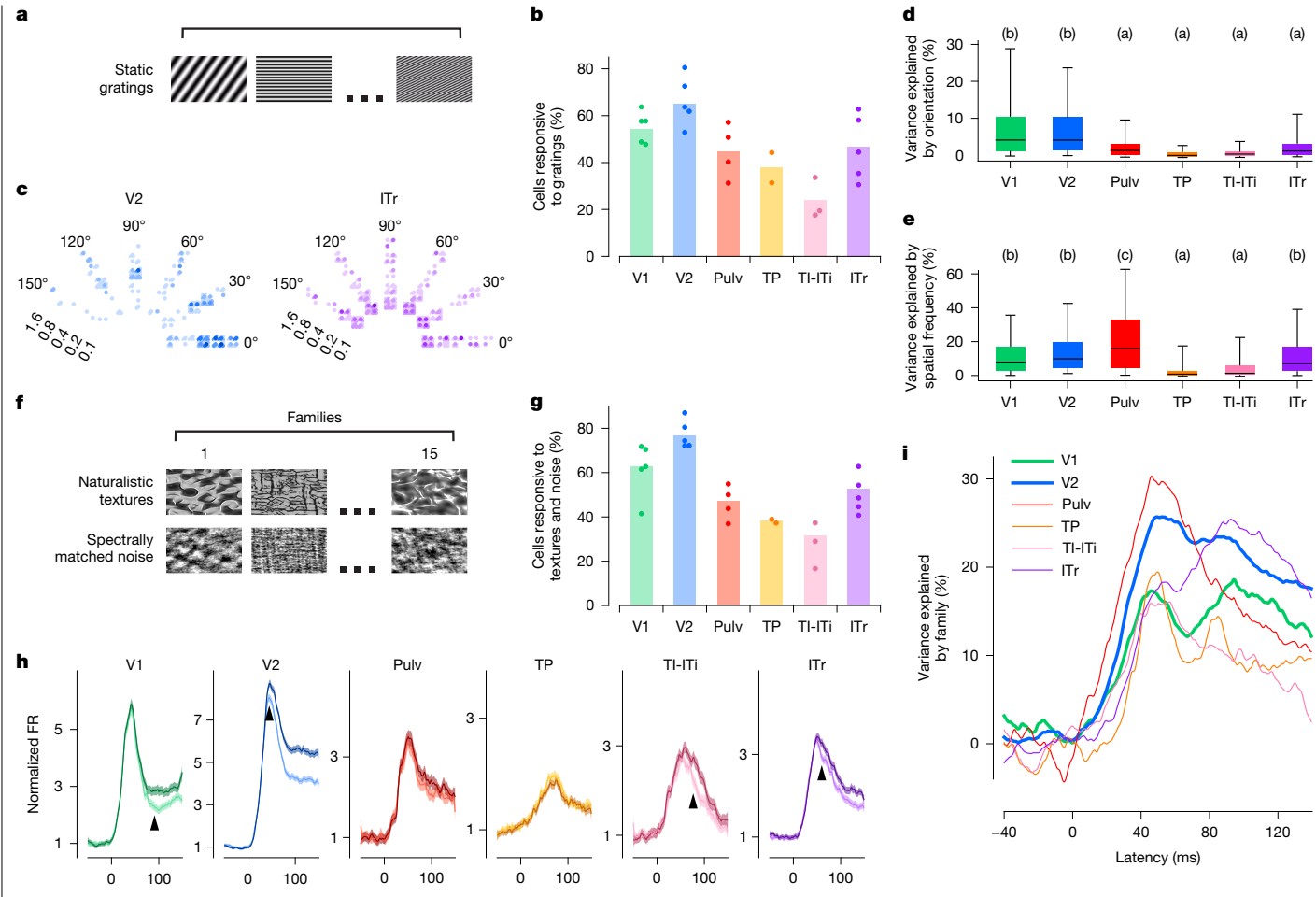

**Fig. 2 | Encoding of orientation, spatial frequency, and texture across tree shrew ventral visual areas. a**, Example frames of static grating stimuli. Stimuli were varied in orientation, spatial frequency and phase, and were interleaved with grey frames. **b**, Percentage of visually responsive cells (see Fig. 1e) that responded to static gratings in individual recording sessions (dots) and averaged across recording sessions (bars). **c**, Responses of a representative V2 and ITr cell to static gratings differing in orientation (represented circumferentially), spatial frequency (represented radially, cycles per degree) and phase (four small quadrants). Each dot represents a single trial; colour intensity represents responses strength. **d**, Percentage of variance of individual cells' responses explained by orientation of the stimulus. Boxes represent 25th, 50th and 75th percentile; whiskers 5th and 95th. Letters in this and subsequent panels indicate Tukey grouping. Tukey analysis ($\alpha = 0.05$) after ANOVA, $F_{5,1106} = 26.791$, $P < 10^{-24}$.

Number of cells: V1 186, V2 500, Pulv 68, TP 79, Tl-ITi 51 and ITr 228. **e**, Same for spatial frequency. Tukey analysis ($\alpha = 0.05$) after ANOVA, $F_{5,1106} = 20.514$, $P < 10^{-18}$ (same cells as **d**). **f**, Example frames of naturalistic texture (top) and spectrally matched noise (bottom). **g**, Percentage of visually responsive cells (see Fig. 1e) that responded to naturalistic texture or spectrally matched noise stimuli in individual recording sessions (dots) and averaged across recording sessions (bars). **h**, Time courses of population responses in each area to naturalistic texture (darker lines) and spectrally matched noise (lighter lines). Black arrows indicate the latency at which the two curves first significantly differed from each other (two-tailed *t*-test, $P < 0.01$). Shaded areas are standard errors of averages across cells. **i**, Percentage of variance in neural activity explained by texture image family (15 classes, see Fig. 2f).

Previous studies in primates have shown that early layers of AlexNet and other DNNs best explain neuronal activity in early retinotopic visual areas, whereas later layers best explain responses in IT cortex[8,13]. We asked whether a similar pattern holds across the tree shrew ventral stream. To test this, we regressed single-cell firing rates against the first 50 principal components of each layer in AlexNet (Methods) and identified the layer that best explained the variance in each cell's response. For one representative cell in V2, AlexNet layer Conv4 best explained its responses (Fig. 4a). Across the V2 population, we found that intermediate layers—specifically Conv4 and Conv5—consistently had greater explanatory power than either early or late layers (Fig. 4b). To compare the explanatory power of different AlexNet layers across brain areas, we calculated the sum across cells within each area of the variance explained by the various AlexNet layers, and normalized these sums by the sum across cells of their explainable variance (Methods). This analysis revealed that early visual areas V1 and

V2 were best explained by intermediate layers—specifically Conv3 to Conv5—whereas anterior areas Tl-ITi and ITr were best explained by the high-level FC6 layer (Fig. 4c). However, the absolute variance explained by AlexNet was lower in these higher cortical areas (Extended Data Fig. 3a,b), consistent with the reduced trial-to-trial reliability of responses to object identity observed in anterior regions (Fig. 3c). One possible explanation is that AlexNet may lack the expressive capacity to fully capture response properties of anterior tree shrew regions, which have been proposed to be multimodal and not exclusively visual[19]

To investigate which feature axes accounted for the most variance in neural responses across areas, we examined how much variance was explained by individual feature principal components from AlexNet layer FC6. In general, earlier principal components explained the greatest proportion of variance in neural responses, with some variability across areas (Fig. 4d). We also analysed how well specific FC6 features could be decoded from population activity in each visual area (Fig. 4e).

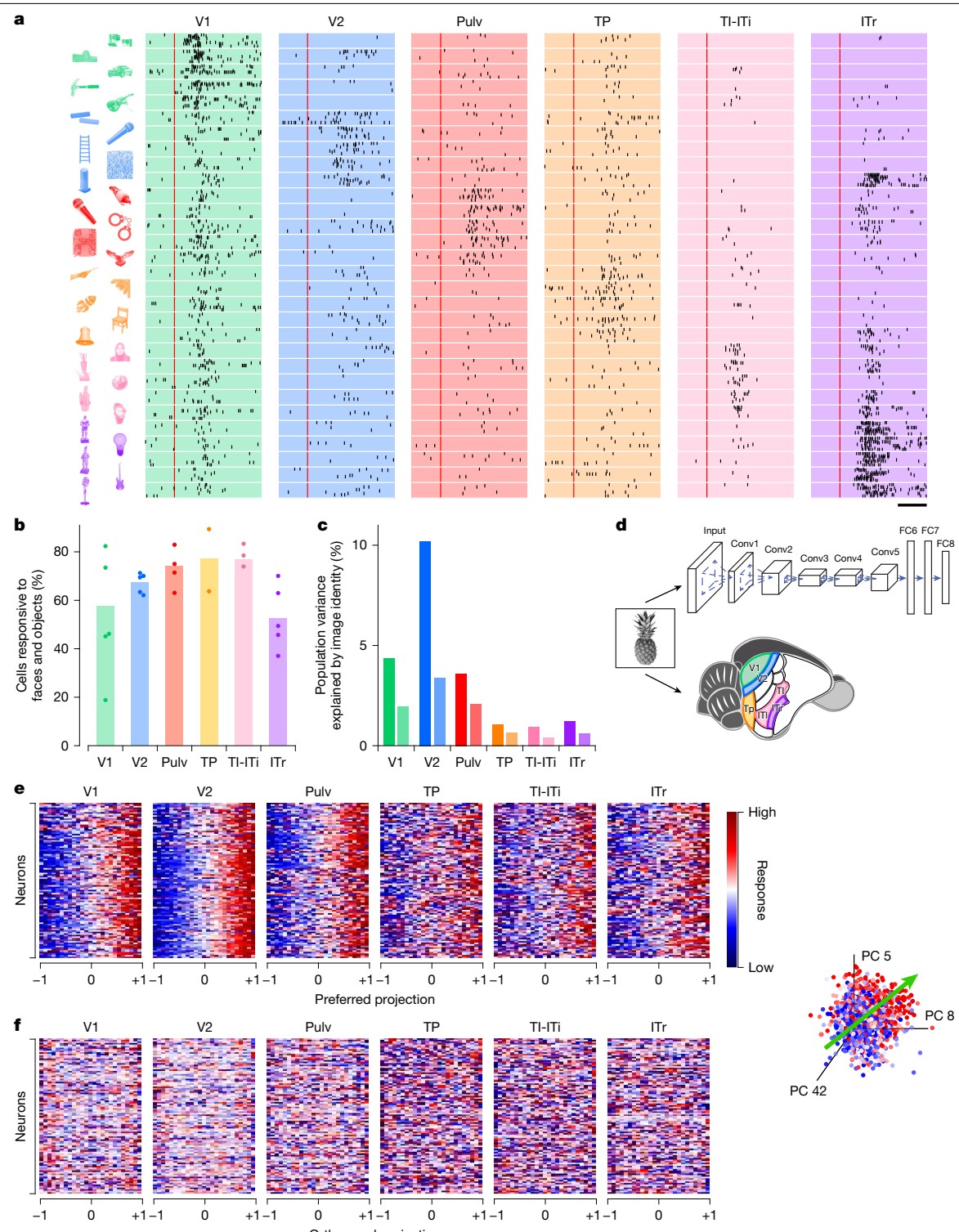

**Fig. 3 | Objects are encoded across tree shrew ventral visual areas through axis coding. a**, Spike raster plots for representative visually active cells from each of the areas in response to six groups of object stimuli, each optimal for one of the cells (stimuli shown on the left). Each dot represents an action potential in one of up to ten presentations of the stimulus; red line indicates stimulus onset. **b**, Percentage of visually responsive cells (see Fig. 1e) that responded to object stimuli in individual recording sessions (dots) and averaged across recording sessions (bars). **c**, Percentage of variance of neural responses explained in each area by object stimulus identity (left bars) and by low-level feature image indices (right bars). **d**, Schematic illustrating the processing of visual stimuli in layers of the artificial neural network AlexNet (top) and in areas of the tree shrew ventral visual pathway (bottom). **e**, Normalized neural responses to object

images for 100 randomly selected cells in each of the six areas as a function of position of that image along the given neuron's preferred axis in AlexNet FC6 space (object space). The x axis is rescaled so that the range [−1,1] covers 98% of the stimuli. Inset, preferred axis (green arrow, Methods) of a representative cell (area V2) in object space. The coordinate axes represent the three AlexNet principal components (PCs) that most align with the cell's preferred axis. Each dot represents an image, colour coded by the strength of the cell's response to that image (blue, low; red, high). **f**, Responses as a function of normalized position along each cell's principal orthogonal axis, that is, the axis in object space orthogonal to the neuron's preferred axis that captured the most variance in AlexNet activations (Methods). Scale bar, 50 ms. Object images in panel **a** used with permission from ref. 8, Springer Nature Limited.

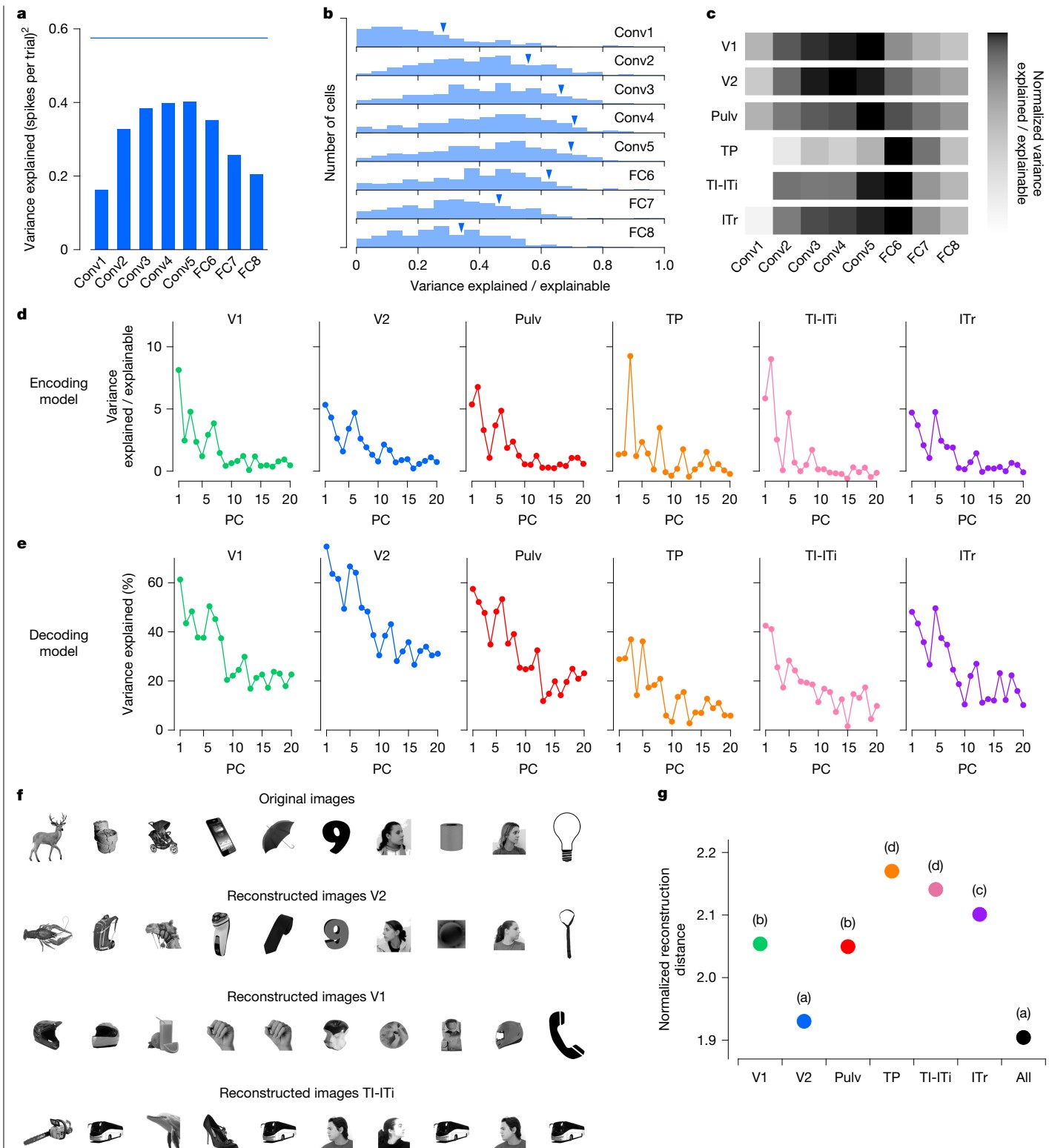

**Fig. 4 | Neural representation of object stimuli in tree shrew ventral visual areas reveals optimal feature decoding in area V2. a**, Variance of the responses of a representative V2 cell explained by individual AlexNet layers. Blue line shows the explainable variance of the cell. **b**, Histograms of explained variance by different layers of AlexNet for responses of visually responsive cells ($n = 602$) in area V2. Blue triangles mark values for the cell from **a**. **c**, Normalized explained variance by AlexNet layers for each tree shrew visual area (Methods). **d**, Variance of encoded neural activity in different areas explained by individual AlexNet FC6 principal components (PCs) as a percentage of explainable variance in that area. **e**, Percentage of variance of AlexNet FC6 features that can be explained by

decoding from the neural responses in different areas. **f**, Ten examples of original images presented to the tree shrew and the images reconstructed from V2, V1 and TI-ITi: that is, the closest images to the predicted responses from AlexNet FC6 from an auxiliary database of images that were not shown to the animal (Methods). **g**, Average decoding distance for each tree shrew visual area between AlexNet FC6 activations predicted from neural activity and actual activations for each image, normalized by theoretical best decoding distance (Methods). Tukey analysis ($\alpha = 0.05$) after ANOVA, $F_{6,11144} = 151.248$, $P < 10^{-184}$. Object images in panel **f** used from ref. 8, Springer Nature Limited.

Again, early principal components were most strongly represented, with decoding performance peaking in V2, substantially higher than in any other region. This finding aligns with the observation that FC6 features explained more variance in V2 than in other areas (Extended Data Fig. 3c). Thus, even though V2 was best explained by Conv4 and Conv5 features, whereas TI-ITi and ITr were best explained by FC6 features, FC6 features were nevertheless better represented in V2 than in these more anterior areas.

Given the strong performance of V2 in decoding AlexNet FC6 features, we next asked whether activity in V2 might be sufficient to reconstruct objects using small neural populations, as has previously been shown in monkey IT cortex[8]. To test this, we used a large auxiliary dataset of 15,901 images, each passed through AlexNet to extract FC6 activations. From activity in each area, we reconstructed an FC6 activation vector and identified the image whose FC6 features were closest to the reconstruction (Extended Data Fig. 3d). To control for cell number, we performed reconstructions using 100 randomly selected cells from each area. Consistent with our results on parameter decoding (Fig. 4e), which were optimal in V2, images reconstructed from V2 closely resembled the original images, whereas images reconstructed from V1 or TI-ITi were notably less accurate (Fig. 4f). To quantitatively compare reconstruction accuracy across areas, we computed the distance between the reconstructed and actual FC6 activation vectors for each image, normalized by the theoretical best decoding distance (Methods). This analysis revealed that V2 had the smallest normalized decoding distances of all areas—indicating the most accurate reconstructions—and matched the performance obtained when pooling neurons from all areas combined (Tukey analysis after ANOVA, $F_{6,11144} = 151.248$, $P < 10^{-184}$; Fig. 4g). These results further underscore the rich yet compact object representation present in tree shrew V2.

Primate IT cortex contains regions composed of neurons that respond maximally to images from specific categories, for example, faces[42–44]. Such category-selective regions can be explained by a normative framework in which IT cortex encodes a general object space—a representational space defined by the first two principal components of the AlexNet FC6 features[8,45]. Within this space, different sectors correspond to distinct object categories, such as faces, fruits and animals (Fig. 5a).

Does the tree shrew visual cortex, like the primate IT cortex, contain regions specialized for representing distinct sectors of object space? To address this question, we projected the preferred axes of all recorded cells onto the same 2D object space (Fig. 5b). In V2, preferred axes were distributed across all four quadrants, whereas in other areas, they were largely confined to quadrants I and III. Given that different object categories are localized to distinct regions of this space, we predicted that individual tree shrew neurons would show selectivity for specific categories. Indeed, analysis of response rasters confirmed that neurons with preferred axes in the face sector were strongly face selective (Fig. 5c). Some face cells also responded to other round shapes, whereas others showed strong selectivity only for faces. In addition to face cells, we identified neurons selective for spiky, elongated objects (quadrant I), round inanimate objects (quadrant II) and spiky animate objects (quadrant IV) (Fig. 5d and Extended Data Fig. 4a,b). However, unlike the modular organization seen in primate IT, we found no evidence for topographic clustering of category-selective neurons within tree shrew visual areas (Extended Data Fig. 4c).

Faces—particularly human faces, which comprised all our face stimuli—are not known to hold special behavioural importance for tree shrews[46]. To confirm that the cells were genuinely face selective, we computed a face selectivity index, defined as the difference between responses to faces and all other objects, for each individual cell (Methods). This confirmed small populations of highly face-selective cells ($t \geq 15$) in most areas starting in area V2, with the highest percentages in TI-ITi and Pulv (Fig. 5e).

Primate IT cortex is highly specialized for object recognition and has long served as a foundation for studying visual form processing. To enable direct comparisons with our tree shrew dataset, we performed large-scale recordings in macaque monkeys using NHP Neuropixels probes. We presented the same 1,593 object stimuli while recording from V2, posterior IT ($IT_{post}$) and anterior IT ($IT_{ant}$) from two monkeys per area (Fig. 6a–c). We found that the explainable variance in responses to complex object stimuli increased along the primate visual hierarchy, from primate V2 to $IT_{ant}$ (Fig. 6d), whereas in tree shrew visual cortex, it peaked in V2 (Fig. 3c). Similarly, image reconstruction performance improved along the primate hierarchy (Fig. 4g), whereas in tree shrews, it was most accurate in V2 (Fig. 6e). In contrast to tree shrews (Fig. 5e), we did not observe strongly face-selective cells in primate V2 (Fig. 6f). As expected, the number of face cells in primate $IT_{post}$ and $IT_{ant}$ was much higher. Notably, in one of the $IT_{post}$ recordings, the probe partially targeted a known face patch, resulting in a higher proportion of face cells.

Last, we asked how well neural populations in the primate and tree shrew visual systems could decode individual face or object identity. To test this, we trained classifiers to decode the identity of either 100 faces or 100 general objects using neural activity from randomly sampled subpopulations within each area (Fig. 6g and Methods). In tree shrews, all areas except TP showed above-chance decoding performance for both faces and objects. When we restricted the analysis to only face-selective cells, face identity decoding improved further. Decoding performance in tree shrew V2 exceeded that in all other tree shrew areas for both face and object identity. By contrast, primate V2 showed substantially lower decoding performance compared with tree shrew V2 (Fig. 6g). Indeed, decoding using tree shrew V2 activity was similar to that of primate posterior IT. As expected, primate anterior IT, which sits at the apex of the primate ventral visual hierarchy, showed the highest decoding accuracy.

A hallmark of the primate ventral stream is gradual emergence of view invariance, raising the question of whether a similar progression exists in tree shrews[8,31,47]. Using DNN models, we computed a predicted view invariance index (Methods) based on responses to the 1,593 object images. In macaques, this predicted index was positively correlated with the empirically measured view invariance index, and both increased along the ventral hierarchy (Extended Data Fig. 5a–c). Applying the same approach to tree shrews, we found no such trend in the model-predicted responses (Extended Data Fig. 5d,e). This absence of a clear progression suggests that view invariance may not emerge in the same hierarchical manner—or may not be captured by current models—in the tree shrew ventral stream. However, direct empirical testing within each area is needed to determine whether view invariance is a core organizing principle of the tree shrew visual pathway, as it is in primates[32] and rodents[48,49]. Taken together, our findings show how visual processing along a series of interconnected areas in tree shrews compares to primates[5,7–9,13,32] and rodents[50–52], highlighting both important similarities and differences (Fig. 6h).

## Discussion

Hierarchical processing is a central principle of object representation in artificial neural networks and in the primate visual system. Here we sought to determine the extent to which the ventral visual pathway of the tree shrew, a highly visual mammal that is one of the closest existing relatives to the primate[53], is also organized hierarchically. Supporting the presence of hierarchical organization, we found that higher-level visual areas showed increased receptive field sizes, longer response latencies, greater selectivity for naturalistic textures compared with spectrally matched noise and higher proportions of single cells selective for faces. However, the tree shrew visual system showed notable deviations from the primate hierarchy, with area V2 performing many functions typically attributed to primate IT cortex. Indeed, area V2 carried the most complete representation of a high-level object space[8] among

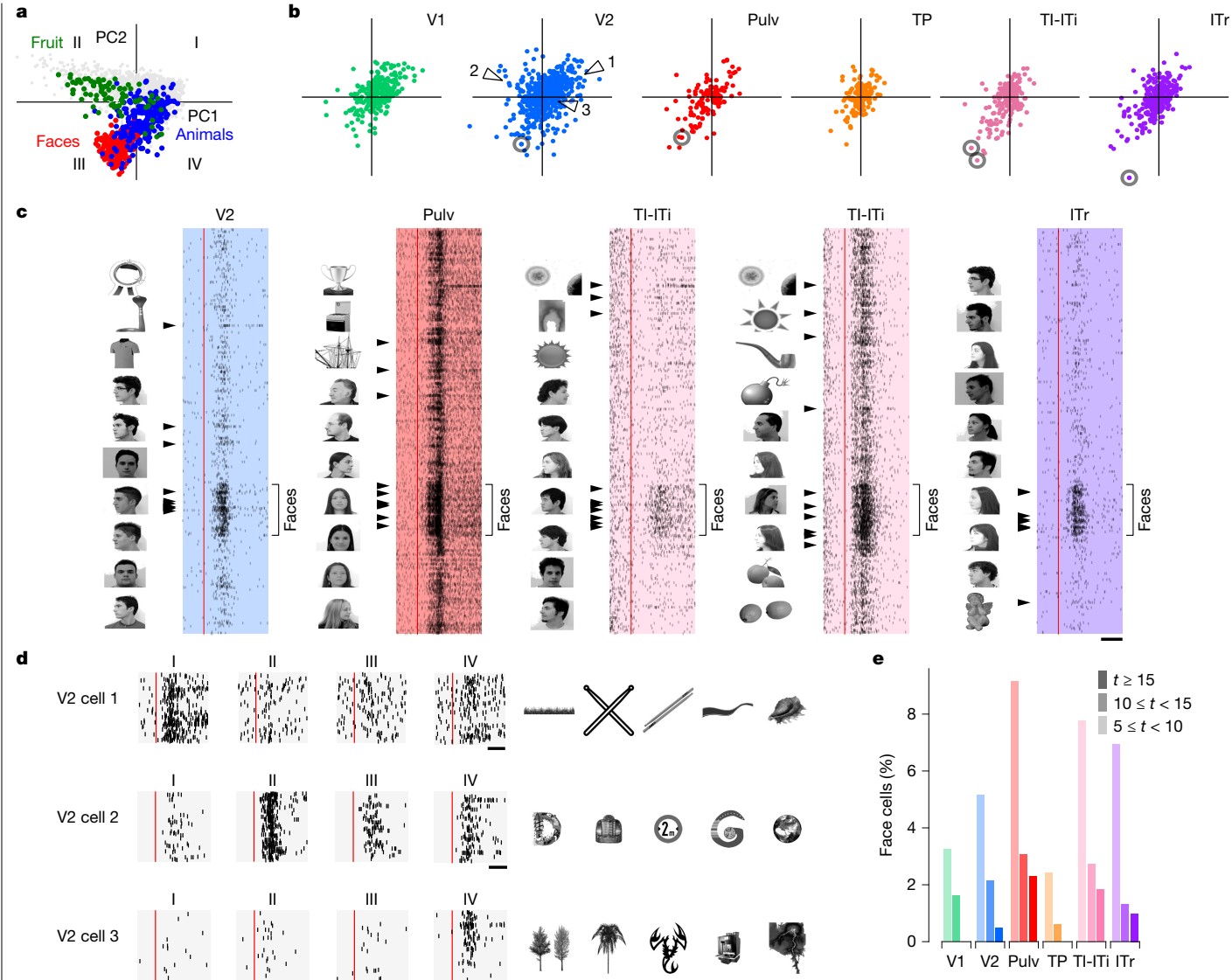

**Fig. 5 | Single cells across the tree shrew ventral stream show selectivity for different sectors of object space including faces. a**, Projections of 1,593 object images onto object space (the first two principal components from AlexNet layer FC6) with images from several categories (faces, animals, fruits) indicated. **b**, Projections of the preferred axes of all cells onto object space. **c**, Raster plots of several representative face-selective cells (circled in **b**) responding to face and object stimuli. The ten most preferred images for each cell are shown to the left of each raster. Arrowheads mark responses to those images. Red lines show stimulus onset. **d**, Raster plots of three representative V2 cells (arrowheads in **b**) with preferred axes in quadrants I, II and IV. Twenty stimuli from each quadrant were randomly chosen to generate raster plots. Right, top five preferred images for each cell. **e**, Histograms of $t$ scores for face selectivity across areas. Scale bars, 50 ms (**c**), 50 ms (**d**). Object images in panels **c** and **d** used from ref. 8, Springer Nature Limited.

all areas, supported the most accurate object reconstruction across the tree shrew visual pathway, and contained strongly face-selective cells similar to those reported in primate IT cortex[9], capable of supporting face identity decoding. These findings suggest a shallower visual hierarchy in tree shrews. An open question for future research is the extent to which tree shrew area V2 encompasses all of the functions of primate IT cortex: for example, whether performance in object recognition tasks can be fully explained by the activity of V2 cells.

As a direct comparison, we performed the same experiments across three homologous regions in the macaque, including V2, IT anterior and IT posterior. This comparison confirmed that tree shrew and primate V2 are functionally distinct: only in the tree shrew do we find a compressed hierarchy, with V2 implementing many of the computations characteristic of primate IT. Our stimulus set was originally tailored for primate object recognition, thereby facilitating direct comparisons to primates. Future work is needed to extend these findings by incorporating further

stimulus sets that include ethologically relevant objects, objects varied in view and depth, and multimodal stimuli. Notably, studies in rats using highly controlled visually morphed objects—designed to match luminance across transformations—led to the discovery of view invariance in rat visual areas laterolateral (LL) and lateral occipto-temporal (TO)[48,49].

Our findings challenge the current focus on modelling mechanisms for high-level vision almost exclusively with deep feedforward networks[54,55]. Computationally, deep feedforward networks assist in the sequential disentangling of image features that are critical for discrimination from orthogonal features such as orientation and size[47,56]. However, it is possible that V2 itself harbours a deep network architecture implemented through local recurrence—for example, a circuit that, when temporally unrolled, could be functionally equivalent to a multi-layer feedforward network but would require fewer neurons to implement[57]. Future work may explore this possibility by analysing the local dynamics of feature selectivity within V2.

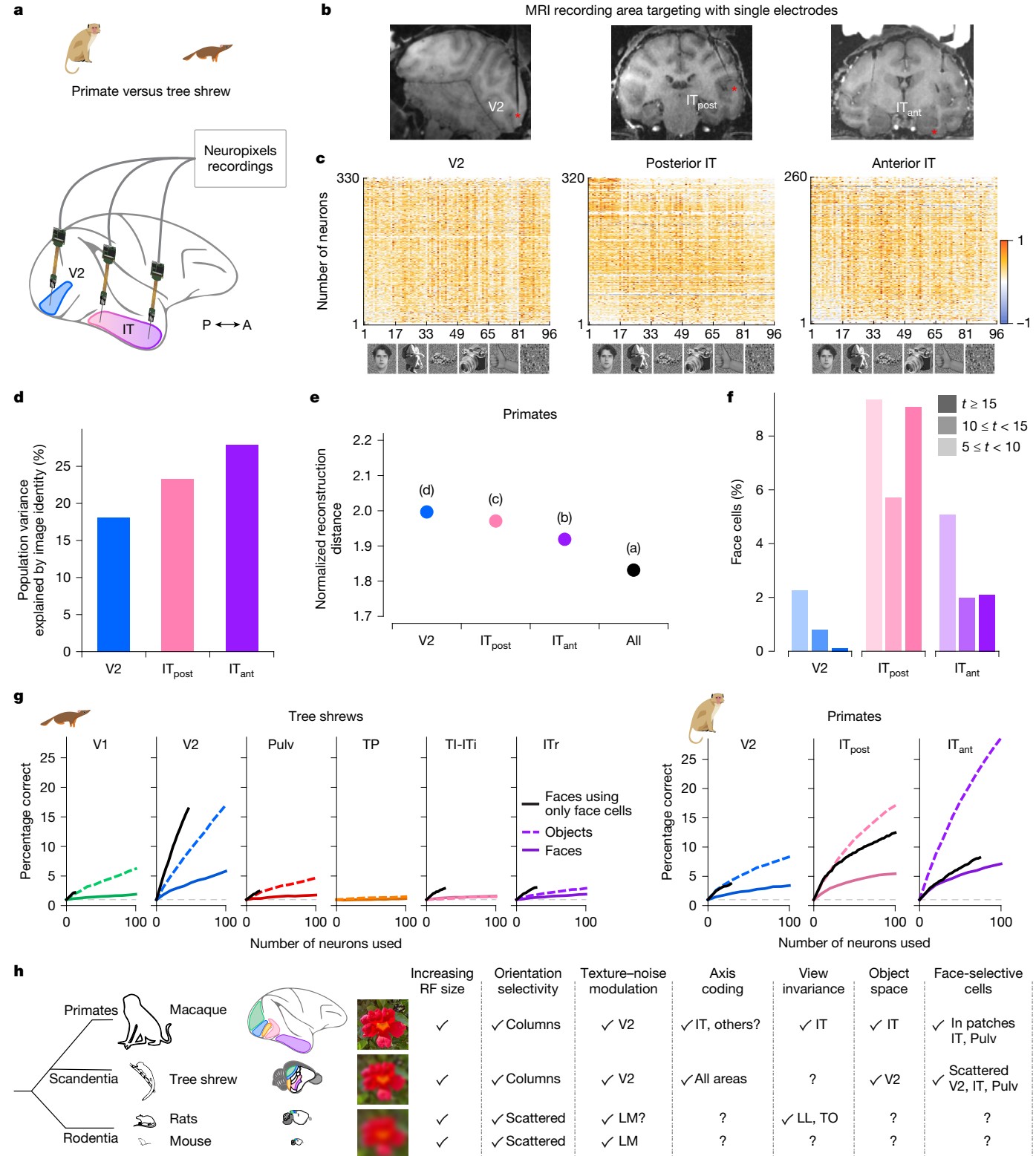

**Fig. 6 | Comparison of object responses between primate and tree shrew ventral visual areas. a**, Schematic of recordings in primate. **b**, Simultaneous Neuropixels recordings from three nodes in macaque monkey cortex. Neuropixels NHP 1.0 probes were inserted into V2, posterior IT and anterior IT cortex. **c**, Responses of cells in V2, posterior IT and anterior IT, respectively (rows), to 96 stimuli composed of faces and objects (columns). Only visually responsive cells were included (two-tailed *t*-test, *P* < 0.05). **d**, Percentage of variance of neural responses explained by object stimulus identity in each area. **e**, Average decoding distance for each visual area between AlexNet FC6 activations predicted from neural activity and actual FC6 activations for each image, normalized by theoretical best decoding distance (Methods). **f**, Histograms of *t*-scores for face selectivity across areas. **g**, Decoding performance for individual object identity (dashed lines) or face identity (solid lines) as a function of number of cells used by the classifier. Note the overlap of the two lines for TI-ITi. Black lines indicate decoding performance for face identity using only face cells (*t* > 5). Dashed grey lines show chance level for object decoding. **h**, Schematic comparing macaque, tree shrew and rodent visual systems. Object images in panel **c** used from ref. 8, Springer Nature Limited.

We did not observe any striking qualitative differences in the complexity of visual processing between area V2 and the more anterior areas in the tree shrew ventral stream (TP, TI-ITi and ITr). This raises the question of what functional distinctions exist between V2 and these anterior areas. One possibility is that the latter are involved in multi-sensory integration, consistent with known anatomical connections to the pulvinar and higher-order auditory cortical areas[19]. Thus, these anterior regions may inherit their visual tuning from V2 without undergoing extensive extra processing, with their primary role being the integration of visual features with features from other sensory modalities.

The finding of face cells in the tree shrew was particularly surprising. The existence of face-selective cells in primates has long been thought to reflect the importance of faces for primate social communication[58]. However, recent evidence argues that such specializations may instead arise from more fundamental principles governing how IT cortex represents a general object space[8,45,59,60]. Facial communication is not known to be ethologically important for tree shrews, which live in isolated monogamous pairs and rely primarily on olfactory cues for social recognition[46]. Thus, the presence of cells selective for human faces in tree shrew visual cortex supports the view that such cells can readily emerge from encoding of general dimensions of image variation, even in the absence of evolutionary pressures related to face-based social communication.

The tree shrew offers exciting advantages as a model organism for studying high-level vision, given its tractability for genetic and virus-mediated circuit approaches and its highly developed visual system. In particular, the tree shrew visual system seems to be more sophisticated than that of the mouse, as evidenced by preferential responses to naturalistic textures, the presence of face-selective cells and the existence of five distinct visual cortical areas (V1, V2, TP, TI-ITi, ITr) showing progressively increasing receptive field sizes and latencies. Our study provides a roadmap for exploring visual circuits in this non-traditional model species and illuminates how brains of different sizes have been adapted for effective representation of the visual world.

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

## Methods

### Experimental model and subject details

All experimental procedures on tree shrews were approved by the Caltech Institutional Animal Care and Use Committee and conformed to local and US National Institutes of Health guidelines, including the US National Institutes of Health Guide for Care and Use of Laboratory Animals.

Three male rhesus macaques (*Macaca mulatta*) were used in this study. All macaque procedures conformed to local and US National Institutes of Health guidelines, including the US National Institutes of Health Guide for Care and Use of Laboratory Animals. All macaque experiments were approved by the UC Berkeley Institutional Animal Care and Use Committee.

No statistical methods were used to predetermine sample size. The experiments were not randomized, and investigators were not blinded to allocation during experiments and outcome assessment.

### Tree shrew experiments

Tree shrews (*Tupaia belangeri*) used in this study ($n = 5$), both male and female, were 6 months to 2.5 years old and weighed between 150 g and 300 g. Animals were singly housed in a 12-h light–dark cycle in the animal room. Their food and water aliquots were given ad libitum.

**Surgeries.** Tree shrews were injected with a preoperative dose of dexamethasone (5 mg kg$^{-1}$, subcutaneously (s.c.)) and mannitol (1 mg kg$^{-1}$, s.c.) to reduce swelling. Animals were anaesthetized with a cocktail of fentanyl, midazolam and dexdomitor (fentanyl 0.05 mg kg$^{-1}$, midazolam 5.0 mg kg$^{-1}$, dexdomitor 0.25 mg kg$^{-1}$, s.c.), shaved and positioned into a stereotaxic frame. Topical lidocaine gel (2%) was applied on the head and ears to prevent discomfort from ear bars and eye lubricant was used to maintain hydration and clarity of eyes during surgical procedures. Levels of anaesthesia, breathing, SpO$_2$ and heart rate were monitored throughout the entire procedure and body temperature was maintained with a heating pad at 37.5 °C. An incision on the scalp was performed and both skin and muscles were retracted. The exposed skull was levelled using the stereotaxic device with respect to bregma and lambda (pitch, roll and yaw). After alignment, locations of the craniotomies for electrophysiological recordings were marked on the skull and a custom stainless steel headplate was secured to the skull using clear C&B Metabond (Parkell). A layer of Kwik-Cast (World Precision Instruments) was added on top the skull and a three-dimensionally printed custom cap was secured to the headplate to protect the brain and keep debris out. The anaesthesia was reversed with an injection of atipamezole-flumazenil (atipamezole 1.25 mg kg$^{-1}$, flumazenil 0.25 mg kg$^{-1}$, s.c.) and the animal was allowed to recover for at least 3 days before following procedures and recordings. One day before electrophysiological recordings in a new brain location, tree shrews were once again anaesthetized and monitored as described above. Using the marked locations on the skull, small (up to 1.5 mm of diameter) craniotomies were drilled and durotomy was performed. Through a small hole situated anterior of bregma, a 32 AWG chlorinated silver wire (A-M System) with a presoldered gold pin was implanted just above the brain surface and cemented to the skull to provide chronic grounding. A drop of silicone oil (30,000 cSt, Aldrich) was added over the holes to prevent brain from drying, a new layer of Kwik-Cast was applied on top of it and the three-dimensionally printed custom cap secured to the headplate. Anaesthesia was reversed and animal was allowed to recover as previously described.

**Electrophysiological recordings.** All electrophysiological recordings were made in awake, head-fixed tree shrews using high channel-count, silicon, Neuropixels v.1.0 probes configured to always acquire from the first 384 electrodes closest to the tip, providing a 3.84 mm of tissue coverage. The reference and the ground contacts on the Neuropixels probes were permanently soldered together. Recordings were made using an external reference configuration achieved by connecting the probe reference to the chronically implanted silver wire on the skull from which conductivity was routinely checked before recording with a multimeter. Each Neuropixels probe was mounted on a three-axis micromanipulator (New Scale Technologies) that was in turn mounted on the underside of a semicircular platform, allowing simultaneous insertion of up to four probes at different angles. Before the first insertion of a probe in a new location, DiI (1 mM in ethanol) was used to coat the shank, allowing subsequent probe track localization during ex vivo imaging. Neural signals were acquired at 30 kHz using OpenEphys software[61]. After the tip of each probe touched the surface of the brain, they were lowered to target at an average speed of 100 μm min$^{-1}$ to avoid damage and let them settle for 15 min after reaching the target depth. Cameras were used to monitor animals during experiments and to ensure a continuative viewing of the visual stimuli presented during neural signals acquisition. After each recording experiment, probes were slowly retracted and immersed in 1% Tergazyme solution to remove tissue and silicone oil residues. For tree shrew Neuropixels experiments, we recorded from six anatomically distinct areas and 5,690 total isolated single units. We recorded from V1 (Bregma −7.5; $n = 695$ units across 5 recordings from 2 animals), V2 (Bregma −6.8; $n = 1,210$ units across 5 recordings from 2 animals), Pulv (Bregma −4.0; $n = 349$ units across 4 recordings from 2 animals), TP (Bregma −6.4; $n = 598$ units across 2 recordings from 1 animal), TI-ITi (Bregma −3.5; $n = 1,421$ units across 3 recordings from 2 animals) and ITr (Bregma −1.5; $n = 1,417$ units across 5 recordings from 2 animals).

**Injections.** To trace the inputs to TP and ITr, intracranial injections were performed as described in the surgical procedure as above. The retrograde tracer cholera toxin subunit β was injected into TP (CTβ-488) and into ITr (CTβ-594) using a pulled glass capillary (World Precision Instruments) and a pressure injector (Micro4 controller, World Precision Instruments), at a flow rate of 50 nl min$^{-1}$. The tracer was delivered at two depths below the cortical surface, 1 mm apart, to ensure adequate spreading. Stereotaxic injection coordinates were based on the Zhou and Ni Tree Shrew brain atlas[62] (TP, anterior–posterior −6.43 mm, medial–lateral ±8 mm, dorsal–ventral −5.5 mm; ITr, anterior–posterior −1.54 mm, medial–lateral ±8 mm, dorsal–ventral −5 mm relative to bregma). Perfusions and histology were performed 7 days following injections.

**Histology.** After electrophysiological recordings or tracer expression, histological verification was performed for all tree shrews. Tree shrews were given ketamine + xylazine and perfused transcardially with 0.9% saline, followed by 4% paraformaldehyde in 1× PBS. Brains were extracted and postfixed overnight in 4% paraformaldehyde at 4 °C. The brains were then transferred to 30% sucrose for cryoprotection and sectioned coronally at 100 μm on a cryostat (Leica Biosystems). Sections were washed with 1× PBS and then incubated for 30 min at room temperature in 4,6-diamidino-2-phenylindole/PBS (0.5 μg ml$^{-1}$) for counterstaining. Sections were then mounted on slides and imaged with an epifluorescence microscope (Olympus VS120). For all representative images, similar results were obtained from at least two independent experiments.

### Macaque experiments

Electrophysiological procedures in macaques followed previously described methods[4,60]. We used Neuropixels 1.0 NHP probes (45 mm long, with 4,416 contacts along the shaft, 384 of which are selectable at any given time) to perform electrophysiological recordings targeted to V2, posterior IT and anterior IT. Data were acquired using the OpenEphys platform, and spike sorting was carried out using Kilosort3. To improve alignment between the guide tube and the probe, we developed a custom insertion system consisting of a linear rail bearing and

a three-dimensionally printed fixture, enabling precise control of the insertion trajectory. Recording sites were selected using magnetic resonance imaging-guided targeting methods as previously described[63]. During electrophysiology and behavioural experiments, monkeys were head-fixed and performed a passive fixation task in a dark room. Visual stimuli were shown on a liquid crystal display monitor (Asus ROG Swift PG43UQ) spanning 26.0° × 43.9° of visual angle. Gaze position was continuously monitored with an infrared eye-tracking system (Eyelink), sampled at 1,000 Hz. Monkeys fixated on a small central spot (0.2° diameter) and received juice rewards every 2–4 s for maintaining successful fixation.

## Visual stimulation

**Visual stimuli presentation.** Visual stimuli were generated and presented using custom Python scripts. Head-fixed tree shrews passively viewed a battery of visual stimuli shown using a ViewSonic monitor (70 × 39 cm, 60-Hz refresh rate, 1,920 × 1,080 pixels). The monitor was centred in front of the animals at a 25-cm distance. Stimuli were presented at 3 Hz, 167 ms of image presentation interleaved with 167 ms of a grey screen. Three classes of visual stimuli were used in each experiment: static gratings, naturalistic textures and noise, and 1,593 objects. In addition, 'local sparse noise' stimuli were used to map neurons' receptive fields.

**Local sparse noise.** The screen was divided into a grid of 4 × 3 squares. In consecutive frames (100 ms), sparse white or black dots (5° square) were presented, one dot in each grid square. The locations of the dots within each rectangle were pseudo-randomly distributed to avoid spurious correlation between distant parts of the visual field[27]. To avoid interference between reconstruction of on and off receptive fields, each presented stimulus frame comprised either all black or all white dots on a grey field. A reduced version of this stimulus (with fewer frames) was used at the beginning of each experiment and analysed immediately to allow placement of 'faces and objects' stimuli in the centroid of the receptive fields for that recording session.

**Static gratings.** We presented full field sinusoidal gratings, varying in orientation (six evenly spread angles), spatial frequency (5 values between 0.1 and 1.6 cycles per degree) and phase (four positions), for a total of 120 different stimulus conditions. Each image was presented five times.

**Naturalistic textures and noise.** We presented images from two subclasses: naturalistic textures and a control set comprising spectrally matched noise. The naturalistic textures images were organized as 15 families of 5 similar images. Each of the 150 images in the stimulus set was presented 5 times. We used two types of visual stimuli similar to ones previously used in primate studies: one set consisted of 15 families of texture images each comprising 5 closely related image samples of the same texture. These images reproduced statistical dependencies found in natural texture scenes[5,64]; a second control set consisted of noise images spectrally matched to each of the texture families.

**Faces and objects.** We presented images from two subclasses: 1,392 objects and animals from http://www.freepngs.com and 201 faces from the FEI database[8], for a total of 1,593 images. Each image was presented ten times. Images were presented at the previously determined centre of the receptive field of recordable cells and sized to cover 20° of the visual field, which covered most of the recorded neurons' receptive fields.

## Data analysis

**Preprocessing and spike sorting.** Neural signals from electrophysiological recordings were preprocessed by subtracting the median calculated within each group of 24 channels from the data to eliminate common-mode noise. The median subtracted data were sent to Kilosort2 for tree shrews and Kilosort3 for macaques[65]. Group median subtraction was applied, followed by a high-pass filter (150 Hz) and then whitening in blocks of 32 channels. The clusters automatically labelled by Kilosort algorithm as 'good' were in turn manually curated by hand and further analysed with Phy2.

**Visually responsive cells.** A cell was deemed responsive to a particular class of stimuli (gratings, textures and noise, or faces and objects) if its average firing rate in the period [0 100] ms following onset of stimuli of that class exceeded the expectation value based on a Poisson model trained on the firing rate in the period [−50 0] ms before onset of all the stimuli of that class. To be included in the 'responsive fraction' in Figs. 2b,g and 3b, a cell's average response had to exceed the baseline by at least five standard deviations. For the faces and objects, the total time elapsed between the first and the last of the ten blocks of visual presentations was so long (~80 min) that stability of responses was a concern. Accordingly, we also preprocessed these data to analyse only those blocks in which the responses were stable for a given cell. For each block, we extracted the average waveform of all the spikes from the given cell and calculated its peak-to-peak amplitude. We then picked the third largest amplitude among the blocks and set an amplitude threshold at 0.6 times this value. We counted for each block the number of individual spikes with amplitudes exceeding this threshold. We calculated the mean and standard deviation of these counts among blocks, and excluded from analysis any block in which the count was more than two standard deviations below the mean. In all cases except Fig. 1e, results are expressed as a percentage of visually responsive cells, that is, of cells that respond to any of the stimulus classes.

**Receptive field analysis.** The receptive field size, amplitude and quality were computed by first calculating a 2D histogram of spike counts at each of 576 locations on the monitor (32 × 8 matrix). We modelled these histograms as a 2D Gaussian peak on top of a constant baseline. To prevent overfitting, the shape of the Gaussian was forced to be circular rather than elliptic. A cell was considered to possess an (ON or OFF) receptive field if the number of spikes within the Gaussian peak exceeded expectation from a null model. Specifically, we calculated the expected number of spikes that would be elicited by (ON or OFF) stimuli within a 10° radius from the centre of the Gaussian under the null model of the baseline as well as the actual number of spikes elicited by stimuli within that same area. The number of standard deviations by which the actual number of spikes exceeded the null expectation was considered the 'quality' of the receptive fields. Only cells with ON or OFF receptive field quality greater than five were considered to possess a receptive field.

**Half-peak latency.** For each neuron, we calculated the average response to all the gratings and texture or noise stimuli as a function of latency after stimulus onset. We found the peak value in this peristimulus time histogram and kept only cells in which the peak exceeded the 99.75th percentile of the Poisson distribution predicted from baseline firing. The half-peak latency of a cell was defined as the latency at which its response first exceeded a threshold halfway between its baseline firing rate and the peak.

**Preferred orientation and spatial frequency.** We analysed responses to gratings in terms of orientation and spatial frequency of the gratings. First, we grouped trials by orientation and fitted a modified Von Mises distribution to the response data for each neuron, in which the orientation space of 0° to 180° was treated as the full period for the purpose of the distribution. The preferred orientation of a cell was the centroid of the fitted distribution. Separately, we grouped trials by spatial frequency. We fitted a Gaussian distribution to the responses in log-frequency space. The preferred spatial frequency of a cell was the

centre of the fitted distribution. In Fig. 2d,e, only cells were included in the count in which the amplitude of the (Von Mises or Gaussian) peak was at least 0.5 times the average firing rate during all gratings responses of the given cell.

**Percentage variance explained.** In Fig. 2d,e,i we plot the percentage of variance in neuronal activity that is explained by various discrete or categorical variables. (We treat orientation and spatial frequency as discrete variables here.) First, we calculate the total variance ($V_{total}$) in neuronal activity across trials for a given neuron. Then we regress the activity onto the categorical variable ($k - 1$ more degrees of freedom, where $k$ is the number of values the variable can attain) and calculate the residual variance ($V_{residual}$). By definition, the explained variance is the difference between total and residual variance, and the plotted percentage is: $100\% \times (V_{total} - V_{residual})/V_{total}$. In Fig. 2i, we separately add up the total and residual variances for all the neurons in an area before normalizing. (That is, we plot the percentage of all the variance in the area that is explained by the variable, rather than the average across cells of the explained variance for each cell.)

**Explainable variance.** To derive an upper bound on the maximum fraction of variance that could theoretically be explained by the DNN, we calculated the 'explainable variance' of the neuronal responses as the split-half reliability of those responses using the Spearman–Brown formula $2\rho/(1+\rho)$ applied to the correlation between the responses to the same image in one half of the trials to the other half[8].

**Preferred axis (AlexNet).** We extracted the activations of the 4,096 units in layer FC6 of AlexNet in response to each of the 1,593 images and performed principal component analysis to reduce the 4,096-dimensional space down to 50 dimensions. For each cell, we calculated which axis in this space captured the largest fraction of the variance in its responses to all but 10% of the images. We then calculated the projection onto the found axis of the remaining images. We repeated this process ten times, each time keeping a different set of images as a test set. This yielded projection values for every image in the dataset. The average of the ten axes found is the cell's overall preferred axis. We defined bins over the projection values and calculated the average response of the cell to all the images in that bin. Each pixel in the matrices in Fig. 3e represents one such average.

**Principal orthogonal axis (AlexNet).** As a control, we took the first principal component of the AlexNet responses and, for each cell, projected it down to the hyperplane orthogonal to that cell's preferred axis. This we call the principal orthogonal axis for that cell.

**Encoding and decoding AlexNet activation from neural activity.** For the encoding analysis of Fig. 4d, we predicted each neuron's image-evoked firing rates from AlexNet principal components using linear regression. Model performance was evaluated on held-out images. The total variance explained by each PC across all neurons within an area was normalized by the total explainable variance across those neurons (estimated by split-half reliability across stimulus repeats). For the decoding analysis of Fig. 4e, we repeatedly selected 100 cells at random from a given area, and used these subsampled cell populations to regress the individual principal components of AlexNet. After repeated sampling, we calculated the average fraction of the AlexNet principal component's variance that was explained by the neural data from a given area.

**Object reconstruction and normalized decoding distance.** Image reconstructions were performed as previously described[8,66]. To generate images that reflect the features encoded in the neural responses, we passed into AlexNet images from an auxiliary database comprising a much larger set of 15,901 images, none of which was previously shown

to the animal. For each stimulus image presented to the animal, the feature vector decoded from the neural activity was compared with the feature vectors of the larger auxiliary stimulus set. We defined the reconstructed image as the image in the auxiliary dataset with the smallest Euclidean distance to the decoded feature vector of the original image.

Given that the auxiliary images used for reconstruction did not include any of the objects shown to the animals (limiting how good the reconstruction can be), we computed a 'normalized decoding distance' to quantify the reconstruction accuracy for each object:

$$\text{Normalized decoding distance} = \frac{|\mathbf{v}_{recon} - \mathbf{v}_{original}|}{|\mathbf{v}_{best\ possible\ recon} - \mathbf{v}_{original}|}$$

where $\mathbf{v}_{recon}$ is the feature vector reconstructed from neuronal responses (obtained by using the Moore–Penrose pseudoinverse to transform the predicted features from neuronal data back into the space of AlexNet layer FC6 activations), $\mathbf{v}_{original}$ is the AlexNet FC6 feature vector of the image presented to the animal and $\mathbf{v}_{best\ possible\ recon}$ is the feature vector of the best possible reconstruction. A normalized distance of one means that the reconstruction has found the best solution possible.

**Face selectivity.** For every cell we quantified its selectivity to faces by calculating the $t$ score between its responses to faces and its responses to the rest of the images. In Fig. 6c, we sorted cells in both macaque IT targets by face selectivity index calculated as FSI = (mean response$_{face}$ − mean response$_{non\text{-}face\ objects}$)/(mean response$_{face}$ + mean response$_{non\text{-}face\ objects}$).

**Face and object identity decoding.** We trained Gaussian naive Bayes classifiers to decode the identity of objects or faces from neural population activity. For each area, we repeatedly sampled a fixed number of neurons (up to 100) and extracted their trial-by-trial spike counts in response to a randomly chosen set of 201 images (objects or faces). Classifiers were trained on 9 of 10 repeats per image and tested on the held-out repeat, with the train/test assignment randomized across repetitions. This sampling procedure was repeated 500 times for neuron subsets and 200 times for train/test splits. Decoding accuracy was defined as the fraction of correctly identified images, and chance performance (1/201) was indicated by dashed lines. Final accuracy curves reflect the mean performance across all resamplings of neurons and repeats.

**Invariance index.** The invariance index was calculated as the mean of the Pearson correlation coefficients between the frontal view and 23 other non-frontal views, averaged across identities. This index reflects the consistency of the neuronal response to the same stimulus presented under different conditions.

**Low-level image features.** To test whether our results could be attributed to neuronal responses to low-level image features, we performed three analyses. First, we calculated 13 different indices for each image, capturing low-level properties such as overall luminance, object contrast and basic shape parameters. For each visual area, we then determined the percentage of the total variance in neuronal firing explained by each index. Second, after mapping the specific receptive field locations for individual cells, we quantified how much variance in each cell's firing rate could be explained by the luminance or contrast within its receptive field following previously described methods[48]. Aggregate results per area were then compared with the amount of variance explained by image identity. Finally, to test whether the shrew visual system was limited by spatial resolution, we divided the 1,593 images into seven groups based on their spatial frequency content and calculated the fraction of neuronal population activity explained by image identity within each group.

## Reporting summary

Further information on research design is available in the Nature Portfolio Reporting Summary linked to this article.

## Data availability

The data that support the findings of this study are available from the corresponding authors (F.F.L. and D.Y.T.) upon reasonable request.

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

**Acknowledgements** This work was supported by a National Institutes of Health Pioneer Award grant no. DP1-NS083063, the Howard Hughes Medical Institute and the Tianqiao and Chrissy Chen Institute for Neuroscience at Caltech. We thank D. Fitzpatrick for help establishing a tree shrew colony and technical assistance; N. Schweers for technical support; F. Luongo for assistance with tree shrew electrophysiology; J. Hesse for assistance with macaque electrophysiology; Y. Shi for assistance with data analysis; and members of the Tsao laboratory, G. Mountoufaris, W. Gonzalez and L. Salay for critical feedback on the paper.

**Author contributions** Conceptualization was performed by F.F.L., D.A.W. and D.Y.T. Software for visual stimulus presentation was written by D.A.W. Formal analysis was carried out by F.F.L. and D.A.W. Experiments were carried out by F.F.L. The tree shrew colony was managed by F.F.L. and J.W. Surgeries were carried out by F.F.L. and J.W. Resources were obtained by D.Y.T. Writing of the original draft paper was done by F.F.L. and D.Y.T. Review and editing of the draft paper was done by all authors.

**Competing interests** The authors declare no competing interests.

## Additional information

**Correspondence and requests for materials** should be addressed to Frank F. Lanfranchi or Doris Y. Tsao.

**a**

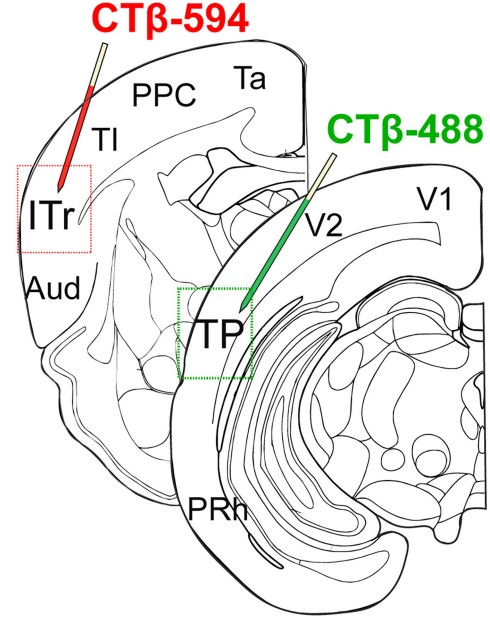

**b**

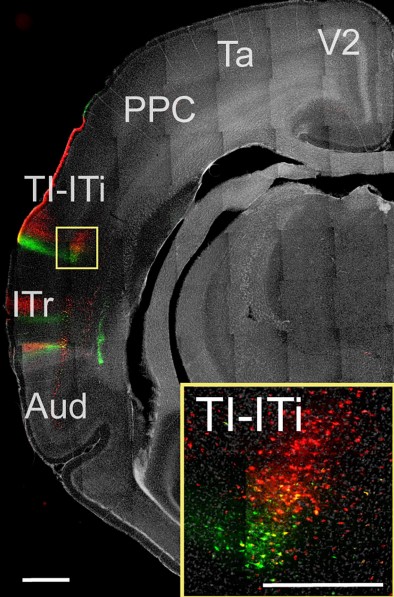
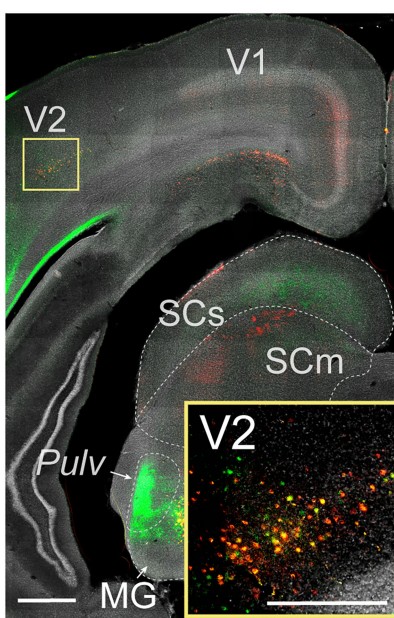

**Extended Data Fig. 1 | Anatomical inputs to intermediate (TP) and anterior (ITr) nodes of the tree shrew ventral pathway.** (**a**) Schematic of injections of retrograde tracer CTβ−488 (green) into TP and CTβ-594 (red) into ITr. (**b**) Coronal histological sections showing retrogradely labeled cells projecting to TP (green) and ITr (red) and counterstained with DAPI (grey). Representative samples out of n = 2 animals. Scale bars: 1 mm / 0.5 mm (insets). Adapted with permission from ref. 62, Springer.

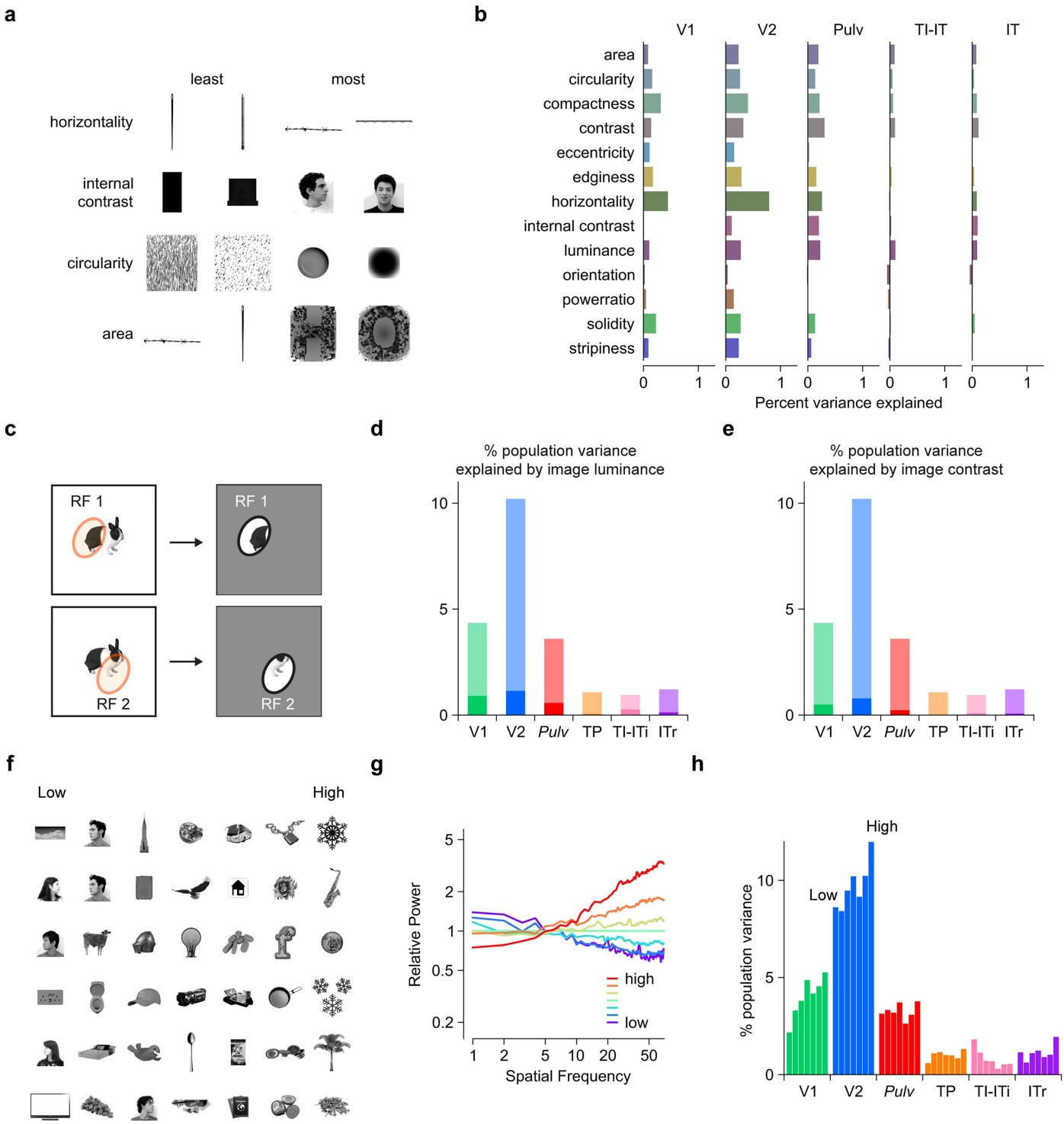

**Extended Data Fig. 2 | Object responses are largely not accounted by low-level features.** (**a**) Examples of the two images with the lowest (left) and highest (right) value for horizontality, internal contrast, circularity and area. (**b**) Histogram indicating the average fraction of variance in the firing rate explained by various low-level image feature indices. (**c**) Schematic of quantification of luminance and contrast impinging on each receptive field. We computed the average luminance and contrast (second derivative of luminance) falling inside the ON and OFF receptive fields of each cell, and average across the two. (**d**) Percentage of variance of neural responses explained by object stimulus identity in each area. Dark bars correspond to the part of the variance accounted for by luminance impinging each receptive field. (**e**) Same, but dark bars correspond to contrast. (**f**) Representative objects with increasing high spatial frequency content from low (leftmost column) to high (rightmost column). (**g**) Power spectrum across groups of images in (a) relative to the middle spatial frequency group. (**h**) Percentage of variance of neural responses explained by object stimulus identity in each area, separated into categories based on spatial frequency. Object images in panels **a**, **c** and **f** used from ref. 8, Springer Nature Limited.

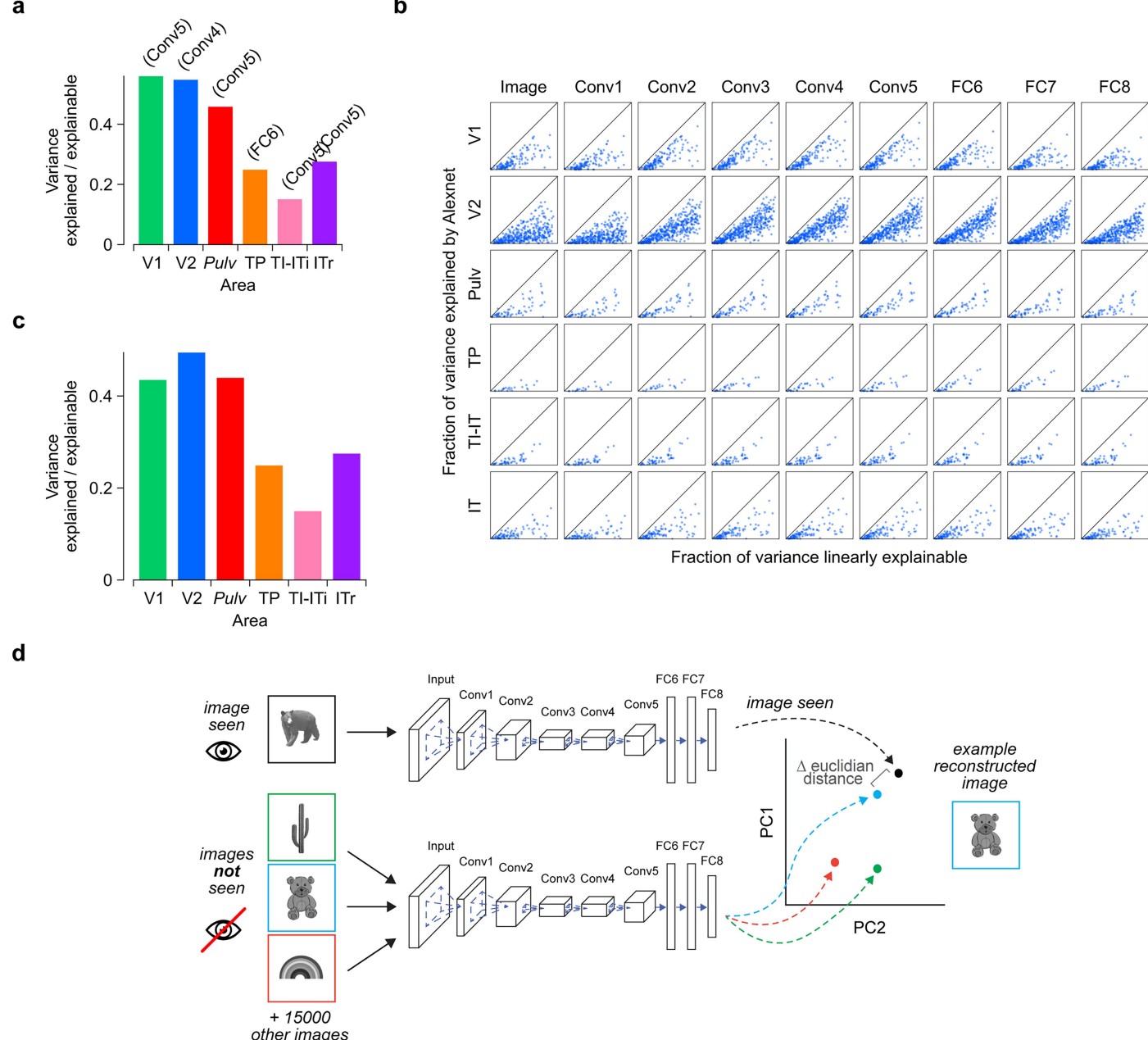

**Extended Data Fig. 3 | Explanatory power of AlexNet and image reconstruction.** (**a**) Aggregate explanatory power of the AlexNet layer that best explained each given area. (**b**) Fraction of variance in the firing rates of individual cells (*dots*) explained by different AlexNet layers plotted against the fraction of the total explainable variance in that cell (*Methods*). (**c**) Aggregate explanatory power of AlexNet layer FC6 over different areas. (**d**) Schematic of image reconstruction approach. Images in panel **d** used from ref. 8, Springer Nature Limited.

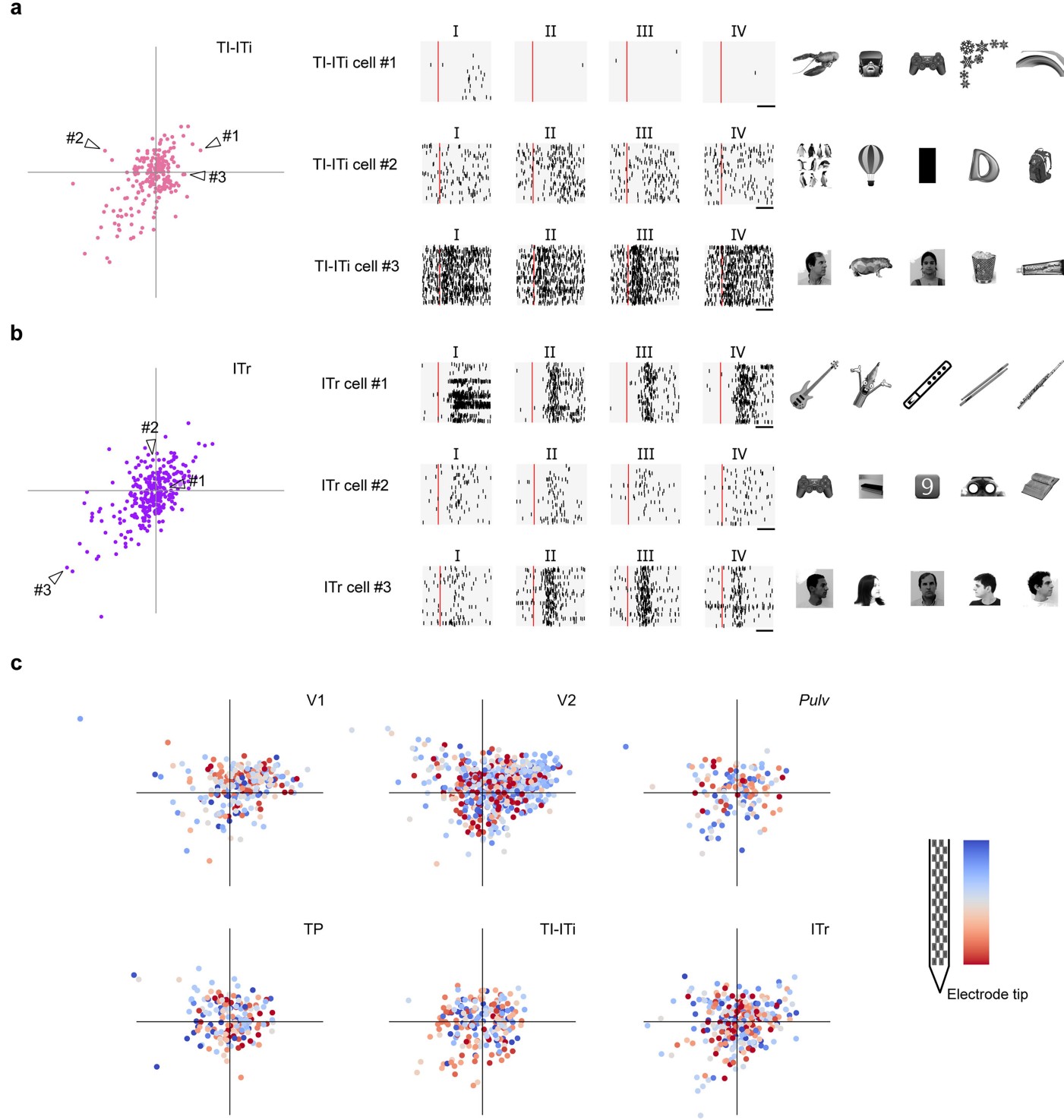

**Extended Data Fig. 4 | Cells selective to different sectors of object space with no obvious topographical organization in object space for each area.** (**a**) Left: Projections of each TI-ITi cell's preferred axis onto the first two PCs of object space (replicated from Fig. 5b). Right: Raster plots of three representative TI-ITi cells from quadrants I, II, and IV indicated by letters; twenty stimuli from each quadrant were randomly chosen to generate raster plots. Scale bar: 50 ms. Top five preferred images for each cell. (**b**) Same for ITr. (**c**) Selectivity of cells in each area as a function of recording depth along the Neuropixels probe. In each of the six plots, each dot represents one cell, the color of the dots indicates the depth at which the cell was recorded (inset, right), and the position of the dot indicates the mean projection of the 10 most preferred images onto the first two PCs of object space. Object images in panels **a** and **b** used from ref. 8, Springer Nature Limited.

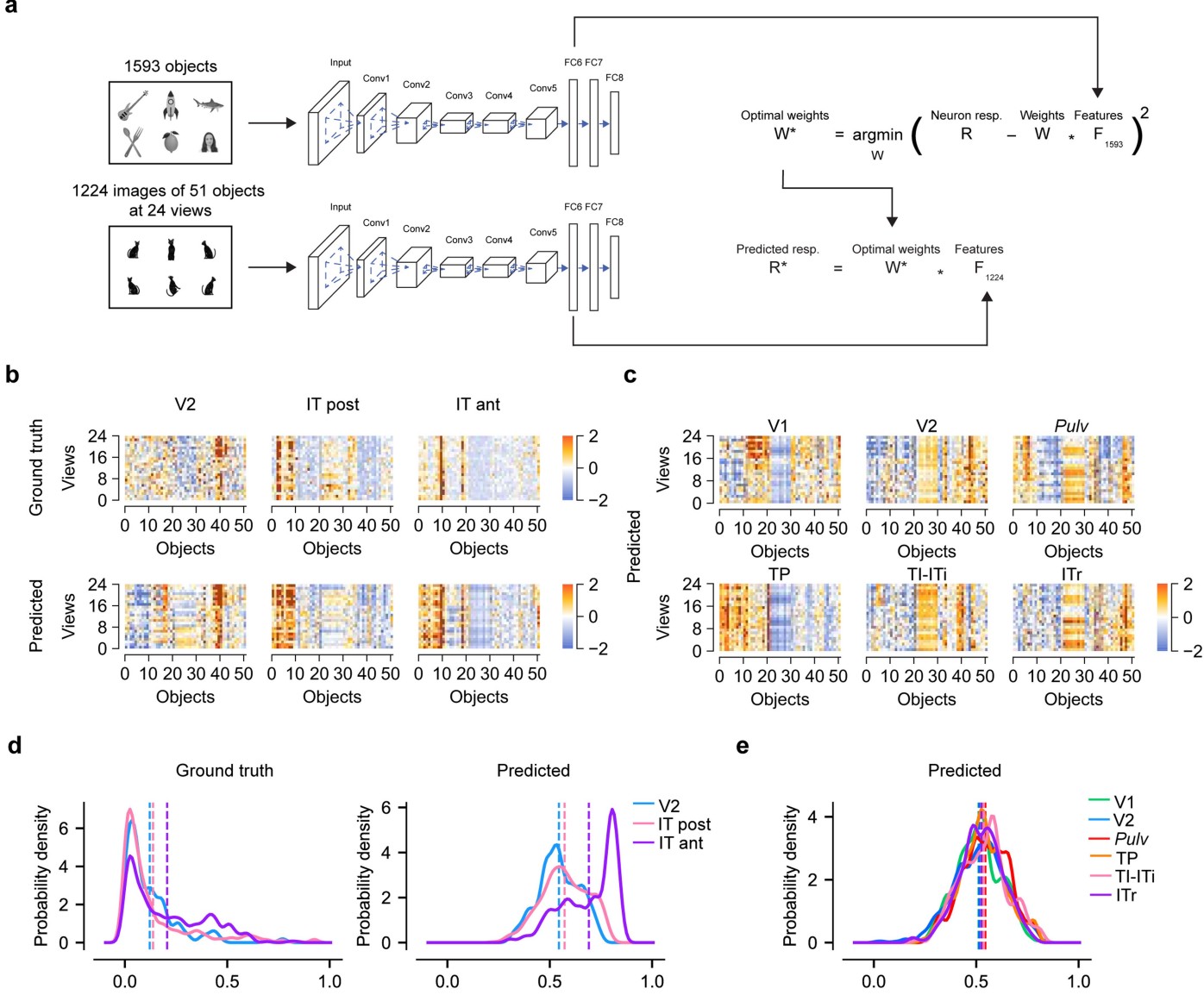

**Extended Data Fig. 5 | DNN-predicted indices of view invariance are similar across all tree shrew ventral visual areas. (a)** Schematic showing workflow for predicting neuron responses for a new set of stimuli. 1593 images were passed through AlexNet (top). Activations in AlexNet layer FC6 were used to linearly predict neural responses evoked by each image when shown to the animal. This yields a weight matrix W that optimally predicts a neuron's response based on the image features F. Next, the weight matrix is used to predict neuron responses to 1224 images consisting of 51 objects at 24 views that were not shown to the tree shrew (bottom). **(b)** Responses of three example cells from macaque V2, posterior IT and anterior IT, to 50 objects (columns) each at 24

different views (rows). Top panel show actual responses, bottom panel shows responses predicted from an AlexNet model built from responses to 1593 images (see Extended Data Fig. 5a). **(c)** Same as **(b)** but for predicted responses of six example tree shrew neurons from all areas. **(d)** Histograms of invariance indices (*Methods*) of macaque V2, posterior IT and anterior IT neurons, calculated from actual responses (left) and predicted responses (right). Vertical lines indicate means. **(e)** Histograms of invariance indices of predicted responses across all tree shrew areas. Object images in panel **a** used from ref. 8, Springer Nature Limited.

# Reporting Summary

## Statistics

For all statistical analyses, confirm that the following items are present in the figure legend, table legend, main text, or Methods section.

| n/a | Confirmed | |
|---|---|---|
| ☐ | ☒ | The exact sample size (*n*) for each experimental group/condition, given as a discrete number and unit of measurement |
| ☐ | ☒ | A statement on whether measurements were taken from distinct samples or whether the same sample was measured repeatedly |
| ☐ | ☒ | The statistical test(s) used AND whether they are one- or two-sided *Only common tests should be described solely by name; describe more complex techniques in the Methods section.* |
| ☒ | ☐ | A description of all covariates tested |
| ☐ | ☒ | A description of any assumptions or corrections, such as tests of normality and adjustment for multiple comparisons |
| ☐ | ☒ | A full description of the statistical parameters including central tendency (e.g. means) or other basic estimates (e.g. regression coefficient) AND variation (e.g. standard deviation) or associated estimates of uncertainty (e.g. confidence intervals) |
| ☐ | ☒ | For null hypothesis testing, the test statistic (e.g. *F*, *t*, *r*) with confidence intervals, effect sizes, degrees of freedom and *P* value noted *Give P values as exact values whenever suitable.* |
| ☒ | ☐ | For Bayesian analysis, information on the choice of priors and Markov chain Monte Carlo settings |
| ☒ | ☐ | For hierarchical and complex designs, identification of the appropriate level for tests and full reporting of outcomes |
| ☐ | ☒ | Estimates of effect sizes (e.g. Cohen's *d*, Pearson's *r*), indicating how they were calculated |

*Our web collection on statistics for biologists contains articles on many of the points above.*

## Software and code

Policy information about availability of computer code

| Data collection | We listed all softwares used in the experiments and for analysis in the Methods section.<br>- OpenEphys for electrophysiology<br>- EyeLink system for eye tracking. |
|---|---|
| Data analysis | We listed all softwares used in the experiments and for analysis in the Methods section.<br>- Electrophysiological data was spike sorted using Kilosort (version 2.5 and 3) and manually curated in Phy( version2, https://github.com/cortex-lab/phy?tab=readme-ov-file).<br>- AlexNet (https://github.com/BVLC/caffe/tree/master/models/bvlc_alexnet original version from Krizhevsky A, Sutskever I, Hinton GE. "ImageNet Classification with Deep Convolutional Neural Networks." Advances in neural information processing systems. 2012.)<br>Custom code written in Python was used for analysis, which is available from the lead corresponding author upon reasonable request. |

For manuscripts utilizing custom algorithms or software that are central to the research but not yet described in published literature, software must be made available to editors and reviewers. We strongly encourage code deposition in a community repository (e.g. GitHub). See the Nature Portfolio guidelines for submitting code & software for further information.

## Data

Policy information about availability of data

All manuscripts must include a data availability statement. This statement should provide the following information, where applicable:
- Accession codes, unique identifiers, or web links for publicly available datasets
- A description of any restrictions on data availability
- For clinical datasets or third party data, please ensure that the statement adheres to our policy

> Data and code supporting the findings of this paper are available on request from the corresponding authors.

## Research involving human participants, their data, or biological material

Policy information about studies with human participants or human data. See also policy information about sex, gender (identity/presentation), and sexual orientation and race, ethnicity and racism.

| | |
|---|---|
| Reporting on sex and gender | N/A |
| Reporting on race, ethnicity, or other socially relevant groupings | N/A |
| Population characteristics | N/A |
| Recruitment | N/A |
| Ethics oversight | N/A |

Note that full information on the approval of the study protocol must also be provided in the manuscript.

# Field-specific reporting

Please select the one below that is the best fit for your research. If you are not sure, read the appropriate sections before making your selection.

☒ Life sciences          ☐ Behavioural & social sciences          ☐ Ecological, evolutionary & environmental sciences

For a reference copy of the document with all sections, see nature.com/documents/nr-reporting-summary-flat.pdf

# Life sciences study design

All studies must disclose on these points even when the disclosure is negative.

| | |
|---|---|
| Sample size | No statistical methods were used to pre-determine sample sizes, but our sample sizes were similar to those used in previous publications in both tree shrews such as Lee KS, Huang X, Fitzpatrick D. "Topology of ON and OFF inputs in visual cortex enables an invariant columnar architecture." Nature (2016) and macaques such as Bao P, She L, McGill M, Tsao DY. "A map of object space in primate inferotemporal cortex." Nature (2020). |
| Data exclusions | No datasets were excluded from analysis but some exclusion criteria for single neurons were applied when appropriate (e.g. when a neuron was not visually responsive to the visual stimulus presented, it does not make sense to include it in the further steps of the analysis). A full and detailed section is included in the Method section under ' Visually responsive cells'. |
| Replication | Results were replicated across at least 2-3 different animals for each experiment independently. |
| Randomization | This is not applicable for this study because of no data grouping. |
| Blinding | This is not applicable for this study because of no data grouping. |

# Reporting for specific materials, systems and methods

We require information from authors about some types of materials, experimental systems and methods used in many studies. Here, indicate whether each material, system or method listed is relevant to your study. If you are not sure if a list item applies to your research, read the appropriate section before selecting a response.

## Materials & experimental systems

| n/a | Involved in the study |
|-----|----------------------|
| ☒ ☐ | Antibodies |
| ☒ ☐ | Eukaryotic cell lines |
| ☒ ☐ | Palaeontology and archaeology |
| ☐ ☒ | Animals and other organisms |
| ☒ ☐ | Clinical data |
| ☒ ☐ | Dual use research of concern |
| ☒ ☐ | Plants |

## Methods

| n/a | Involved in the study |
|-----|----------------------|
| ☒ ☐ | ChIP-seq |
| ☒ ☐ | Flow cytometry |
| ☒ ☐ | MRI-based neuroimaging |

## Animals and other research organisms

Policy information about studies involving animals; ARRIVE guidelines recommended for reporting animal research, and Sex and Gender in Research

| | |
|---|---|
| Laboratory animals | Five tree shrews (Tupaia Belangeri) aged 6 moths to 2.5 years old and three male rhesus macaques (Macaca mulatta) aged 5-15 years old were used in this study. |
| Wild animals | The study did not involve wild animals. |
| Reporting on sex | This study was conducted using male and female tree shrews. Only male macaques were used. |
| Field-collected samples | The study did not involve field-collected samples. |
| Ethics oversight | All procedures conformed to local and US National Institutes of Health guidelines, including the US National Institutes of Health Guide for Care and Use of Laboratory Animals. All experiments were performed with the approval of the Caltech and UC Berkeley Institutional Animal Care and Use Committee. |

Note that full information on the approval of the study protocol must also be provided in the manuscript.

## Plants

| | |
|---|---|
| Seed stocks | N/A |
| Novel plant genotypes | N/A |
| Authentication | N/A |

