## [Peer Review File · Nature]

A compressed hierarchy for visual form processing in the tree shrew

Corresponding Author: Professor Doris Tsao, Doris Y. Tsao and Frank F. Lanfranchi

This file contains all reviewer reports in order by version, followed by all author rebuttals in order by version. Parts of this Peer Review File have been redacted as indicated to remove third-party material.

Version 0:

Reviewer comments:

Referee #1

(Remarks to the Author)

The manuscript by Lanfranchi and colleagues describes the results of a creative approach to comparative analysis of visual systems across mammals. The authors used Neuropixels probes to examine visual responses in many tree shrew brain areas to a large stimulus battery. The authors find that cells in area V2 exhibit selectivity for complex objects that is generally not found in area V2 of the primate. The analysis and basic recordings are well done. The conclusion hinted at, that animals with fewer visual brain areas must have processing networks that are less deep or differently constructed (more recurrency than feed-forward, a possibility raised by the authors) is very interesting and would appeal to the broad readership of Nature. The figures are clear and easy to interpret.

I have two major concerns that I believe the authors can address.

First, I wonder if the results follow in a trivial manner from the lower spatial frequency selectivity of the tree shrew visual system as compared to the primate visual system. The stimuli being shown have high spatial frequency characteristics, and clearly cannot be reconstructed as well they could be from recordings from the primate visual system. If one were only going to reconstruct low spatial frequency information, would the hierarchical nature of the primate visual system be less pronounced? This concern applies to the main result of the paper (the early emergence of object-decoding responses already in V2) as well as to the conclusions about view invariance in Extended Data Figure 4. I imagine that other readers may have the same question, so it would be important to address this issue in the text and figures rather than just in a response to the reviewers.

Second, the primate visual system is the comparative anchor of the narrative, yet comparable data are not shown. I realize that the authors likely do not have data from many of these putatively homologous areas, but do the authors at least have data from primate V2 and IT that can be used in even a rough comparison? It would make the paper much stronger to have more than a by-narrative-only comparison (the data in Expanded Data Figure 4 is only for IT) of what we expect V2 to do under the same stimuli.

With those criticisms mentioned, I think the paper is creative and brings a quantitative edge to comparative studies across mammals that is paradigm-shifting and will appeal to the Nature readership.

Small items:

Line 70: reword, sounds like the primates are moving instead of the reader's attention

Line 111: "responded to any of the classes of visual stimuli we tested". There must be some statistical definition of "responded" yes? Please indicate.

Line 303: "A hallmark of the ventral stream" -> "Another hallmark" (phrase was already used in line 258)

Line 611: is there a citation that can be provided for the explainable variance?

Line 639: the paragraph is a little brief, please expand

Extended Figure 1b: What is the counterstain of the tissue? Please include its name in caption and procedures in methods.

Discussion: One idea the authors may consider adding to the discussion is the possibility that the tree shrew's visual system may be configured to process additional stimuli that were not in the stimulus set used by the authors. For example, if the authors had used stimuli with more temporal structure or binocular disparity, they might have observed a more hierarchical organization of optimal reconstruction. The paper reads as though the authors believe the chosen stimulus set can reveal all we need to know about object recognition, but this is really an assumption. It is clearly an interesting stimulus set to examine, but it might be really tailored to primate object recognition (something we can relate to as humans and well represented by IT) rather a general representation of stimuli important to all mammals. I don't think this issue detracts from the paper or need to alter the primary narrative, but I think noting it would add depth to the space of interpretations.

Referee #2

(Remarks to the Author)

In this manuscript, the authors explore whether the anatomical hierarchy of visual areas that, in the tree shrew, is the homologous of the macaque ventral stream harbors an increase of complexity of tuning and encoding properties, with respect to the image space, that would make it homologous to the monkey ventral stream also at the functional level. The authors record the activity from hundreds of neurons in 6 different visual areas of the tree shrew, probing their responses with a rich battery of stimulus conditions, including gratings, textures and objects. They find that some tuning properties follow a monotonic trend along the anatomical hierarchy (e.g., orientation tuning decreases), while other properties (such as tuning for visual objects, ability to discriminate face identity, etc) peak already in V2, rather than in the deepest regions of the pathway. Hence the authors' conclusion of a compressed processing hierarchy in this species.

I found the scientific question very interesting and the study well crafted, although I believe it has two major weaknesses. At the data analysis level, the authors have not explored the extent to which the trends they found can be explained by lower-level areas (V1 and V2) being more sensitive to low/middle level features (such as luminance, contrast, aspect ratio, surface area of the objects). At the experimental design level, they did not include in their stimulus sets any conditions to probe the invariance of the representations, which, among all properties that can be studied along a ventral-like stream, is not only the most relevant but also the one that can be measured more rigorously. Hard to make definite claims about whether an anatomical progression behaves as an object-processing pathway unless invariance is systematically assessed.

I can say this by direct experience, given that I embarked in a journey that is very similar to the one of the authors, having tried to establish whether the rat visual cortex contains a functional object processing hierarchy akin the ventral stream. Full disclosure here: I am Davide Zoccolan, the author of several studies on the rat ventral stream that addressed very similar questions to those addressed here by the authors. I prefer to disclose my name, since many of the criticisms and suggestions I have listed below derive from my very direct experience in dealing with similar data and questions. So, I believe that all my comments will be more understandable if the authors know where they come from.

Connected to this, another general criticism I have for the authors is that they have not made any attempt to compare their findings with any species other than monkeys. Rodent studies are almost fully ignored. This is unfortunate given the evolutionary position of tree shrews (somewhat in between rodents and monkeys) and the efforts that have been made to understand whether rodents possess functional processing hierarchies similar to those of primates.

MAJOR ISSUES

1) Page 4, Line 195-197 "After V2, the explainable variance in responses to these complex object stimuli decreased dramatically ... This contrasts with the primate, where, e.g., ~80% of variance of IT cell responses to a large image set can be explained by image identity. Overall, these results suggest that even though anterior tree shrew areas were relatively more responsive to complex objects than to gratings, their responses to these complex object stimuli were not highly reproducible from trial-to-trial". This result has not to do only with the reproducibility of the response but also with how similar (in units of variance) the mean responses to each stimulus are. So, the conclusion is that the discriminability of individual object conditions decreases from V2 onward. But this finding, without further analysis, does not mean, per se, that deeper temporal areas are less good at encoding high-level visual information. Because of two reasons.

First, when stimuli as complex as objects are presented, there is a very large number of low-to-middle level features that may systematically vary across the objects. For instance, the luminance and contrast that impinge upon the RF of each unit. In our previous study of the rat ventral stream (Tafazoli et al., 2017), we quantified these properties and referred to them as effective RF luminance and contrast. In that paper, we carried out an analysis similar to the one of the authors, but based on the mutual information $I(R;S)$ between stimulus identity (S) and response (R) and we found a very peculiar trend. $I(R;S)$ decreased sharply along the ventral stream (being maximal in V1), similarly to what reported here by the authors. But we discovered that the reason was that units in earlier areas carried way more info about low level features of the objects (luminance and contrast). In other words, V1 units discriminated better among object conditions simply because they were more sensitive to the fact that some objects were brighter or larger or (conversely) darker or thinner. By comparison, LL, the deepest rat ventral area, carried way less info about such low-level properties. We found that this was the reason at the root

of its lower I(R;S). But, once we separated the part of the mutual info about low level properties (e.g., luminance; white bars in Fig. 3B of Tafazoli et al) from that of higher-level features (colored bars in Fig. 3B), it turned out that the latter was equally large in all areas. Therefore, LL carried the largest percentage of higher order info about stimulus identity than all other areas (Fig. 3C). Interestingly, we have shown that this same behavior can be found for the mutual info conveyed by the units of deep nets (VGG family) about luminance and contrast (Muratore et al., 2022).

Looking at the examples of stimuli used by the authors, it seems that they vary substantially in terms of object area, contrast, luminance etc. So, their stimulus set lends itself to the same possible confound of our stimulus set, given the analysis they applied. In other words, the decrease of explained variance across the shrew ventral stream could possibly reflect a decrease of sensitivity to low level properties. But pruning such info would be a key functional feature of a ventral pathway. So, it is essential that the authors control for that. Formally, the best way to do it would be to compute the mutual info between stimulus and response and then break it down in the low-level and higher-level components, as we did in our study, by computing conditional mutual info. The methods section of our paper explains in details how this can be done, but dealing with mutual info analysis applied to neuronal data is always technically tricky. However, the authors could perhaps look at simpler ways to do this analysis: for instance, compute the amount of variance explained by luminance and contrast and then by aspect ratio of the objects or surface area. It would perhaps be enough to regress the responses to these continuous variables and compute the R2. Care should be taken however to quantify how much contrast and luminance entered the RF of each specific unit (again this is described in the method section of our paper). Without such control, it would be unreliable to conclude that there is not a hierarchical increase of object information along the ventral stream of the tree shrew.

2) Page 5, lines 224-228 "... early visual areas V1 and V2 were best explained by early layers Conv3 to Conv5, whereas ITi and ITr were best explained by FC6 (Fig. 4c). However, the absolute explanatory power of AlexNet was lower for the higher cortical areas ...". The comparison between relative and absolute fraction of variance explained by Alexnet layers in each area is very interesting. One could interpret these findings in different ways, but one possibility could be the following. Tuning in higher areas result from a larger number of nonlinear hierarchal transformations and, as such, is much harder to model, even with the latest stages of a deep net. In addition, in these areas, a big chunk of low-level visual info may have been lost, which is the easiest to model. Hence, again, the difficulty of capturing a lot of variance (in absolute terms) in the responses of neurons in the highest visual areas. Yet, the deepest Alexnet layers are the best at modeling these neurons, which means that, in relative terms, a deep functional hierarchy is required to capture some of the tuning of these cells.

Conversely, in V1/V2, responses may be dominated by a strong sensitivity to low level features, less nonlinear transformations have accumulated, and, therefore, overall, regardless of the Alexnet layer used to regress the neuronal data, the absolute amount of explained variance will be higher than in deeper visual cortical areas. Yet, in relative terms, the larger amount of variance will be explained by early-middle layers, given that the complexity of shape representations in V1/V2 is not that large.

In other words, these results are not necessarily in contrast with the existence of a functional hierarchy where visual features' combinations of increasing complexity are progressively encoded. I believe the authors should consider and discuss this scenario and try to test it, ideally by measuring how much tuning for low-level features is present in the different areas.

3) Page 5, lines 232-233 "earlier PCs explained more variance in neural responses, with some variability across areas". Related to my two previous points: this finding could suggest that earlier PCs of the Alexnet representation could be related to lower/middle level visual properties (local luminance, contrast, aspect ratio, surface area of the objects) and that visual neurons are very sensitivity to such low-level attributes. The authors could perhaps measure these low-level properties across their stimulus set and see how well they correlate with the first few PCs of the Alexnet representation.

4) Page 5/6, lines 242-256 "...we asked whether activity in V2 might be sufficient to reconstruct objects using small neural populations ...". I was not able to understand this analysis and, therefore, I cannot comment on the results of the image reconstruction. I urge the authors to expand considerably the Method section that describes the procedure, ideally also adding a supp figure to guide the reader.

5) Page 6, lines 258-301, i.e., Analysis of Fig. 5. Also: page 7, lines 343-346. Again, unless some control is done about the possible encoding of low level features by V2, I am not convinced about the conclusions of the authors concerning this analysis. This 2D projection on the 2 first PC of the representation in Alexnet FC6 sounds very much to me like a projection on a subspace encoding quite low-level features. Although the authors do not mention this, it is sort of implicit in the names they gave to the axes of the space: stubby vs. spiky and animate vs. inanimate (which I would more appropriately call squared vs. roundish perhaps) are properties that correlates strongly (e.g.) with object surface area and, therefore, aspect ratio and luminance. This subspace reminds me of a result we found in monkey AIT (Baldassi et al., 2013), where the first PC that best explained the selectivity of the recorded neurons was a very low-level property: object surface area (basically, the stubby vs. spiky axis of the authors). In that study, we did take this as a sign of the (surprising) fact that IT neurons, despite being so high level, still carry substantial info about stimulus properties that are quite basic and low-level, as lower-level areas are supposed to do. But this is of course not the whole story - because what IT should do (as shown in many studies by DiCarlo lab for instance) is to encode also higher order features (beyond these first two PCs of Alexnet fc6), which lower areas do not encode. In this sense, in my opinion, the fact that neurons in V2 spanned all the quadrants in the 2 PC space is not necessarily a sign of a higher level representation, as compared to deeper areas. Actually, I would expect that tuning in deeper areas is accounted less well by these first 2 PC - that neurons occupy less territory in the 2 PC space (exactly because those neurons now encode features that go beyond the two basic axes stubby vs. spiky or animate vs.

inanimate).

In V2, there could be a very good representation of faces, simply because what V2 neurons encode are low level attributes that faces happen to all share: similar aspect ratio, object surface size, contrast and luminance. One way to demonstrate that there is a real encoding of faces would be to play with those faces and transform them in various ways, so as to test invariance for face encoding. This is exactly the issue we encountered in analyzing the rat ventral stream (Tafazoli et al., 2017). Rat V1 neurons, given their sensitivity to low-level attributes and given that objects of the same category (even despite of transformations) shared low level features, were the best at encoding object identity, even across view changes. Only after carefully matching the mean luminance (across transformations) between the objects to be decoded, we found the signature of a hierarchical increase in the ability of visual areas to encode object identity in an invariant way. Critically, a demonstration of this kind is currently missing from this study. One way to achieve that would be to run a face categorization analysis in which faces are pitched against objects with aspect ratio, surface area and luminance as similar as possible to those of the face class. I believe an analysis of this kind is necessary.

6) Last paragraph of the Results: invariance analysis. I am sorry, but the authors cannot claim that they tested the invariance of the representations across the shrew visual areas. As they acknowledged, their stimulus set did not contain conditions (i.e., multiple views of the same objects) to perform this test. What they have done here (resorting to modeling with Alexnet) is not acceptable as a test of invariance. One cannot simply use the linear regression model with Alexnet units to simulate how the actual neurons would have generalized their responses across transformations. There is no guarantee that the model will be so faithful to allow measuring invariance without recording actual neuronal responses. Every article that has measured invariance in the monkey and, more recently, in the rat ventral stream, has made sure to have carefully designed conditions to do so.

I believe that the authors should not claim they have tested invariance and remove reference to this test in their Results and Discussion. Unless they can add a new set of recordings, even perhaps from just a few checkpoint areas along the hierarchy, to really measure invariance. If they have the possibility to do so, they should pay, again, a lot of attention to possible confounds yielded by sensitivity to low-level visual properties. In our study on the rat ventral stream (Tafazoli et al., 2017), the lowest areas (V1 and LM) were actually the more view invariant, unless the decoding analysis was restricted to pairs of visual objects that were well matched in terms of luminosity across transformations.

IMPORTANT ISSUES

1) Page 3, lines 120-123. "Cells with ON and/or OFF receptive fields were clearly present in all areas except TP (Fig. 1f). Surprisingly, this included the two most anterior areas TI-ITi and ITr...". Fig. 1f is far from ideal to understand how the RF structure changes across the areas. I think it would be more informative to plot 3 bars for each area: one with the fraction of RFs with ON regions, one with the fraction of RFs with OFF regions and one with the fraction of RFs with both ON and OFF regions.

However, it would be even better to carry out an analysis of the structure of the RFs to better quantify: 1) their complexity; and 2) their sharpness. One way to do so would be to measure the contrast of the RFs and to count the number of distinct lobes in the RFs as we did in (Matteucci et al., 2019; Matteucci and Zoccolan, 2020). Please refer to Fig. 5 in Matteucci et al 2019 for further details. Performing such quantification would allow to better support a statement like: "Surprisingly, this included the two most anterior areas TI-ITi and ITr; in contrast, corresponding areas in the anterior primate temporal lobe show largely spatially invariant responses".

In fact, unless one considers the really highest stages of the monkey ventral stream (anterior IT), it is not so surprising for a middle-to-high level visual area to yield some sort of RF using reverse correlation. Even if the stimulus/response relationship is nonlinear (e.g., because of spatial invariance), visual neurons often retain a linear component of their responses and such linear residue may show up as a light or dark region when mapping RFs with sparse or dense noise. The more important question is whether these RFs are structured or not. In V1, simple cells with linear stimulus/response relationships will show a rich structure with multiple, crispy lobes (both ON and OFF). Higher order, more nonlinear units will display fainter RFs with a lower number of distinct lobes. This is what we found in the comparison between V1 and LL (one of the highest stages of the rat ventral stream) in Matteucci et al 2019. I believe it would be important to carry out this sort of quantification also here.

2) Page 4. Line 160-161 "V2 had the largest proportion of cells responding to the texture and/or noise stimuli". This quantification of explained variance is of course informative about the ability of cells in a population to differentiate among textures but it does not provide a direct measure of how discriminable the textures are based on the population activity. Also, I believe that Fig. 2i show the mean (or median?) of the explained variance across the population, so it does not tell how well, collectively, the neuronal ensemble is encoding the textures. So, if the goal is to support a statement like: "the observation that activity in V2 encoded texture family identity earlier than that in V1", I would recommend carrying out a real decoding analysis. As a reference, the authors could take a recent study of textures encoding in mouse visual cortex (Bolaños et al., 2024). As done in this paper, they could use a d' index to measure the discriminability of texture vs. noise at the single cells level and then report a population statistics. They could also carry out a population decoding analysis using a binary classifier probing how well each population can discriminate between texture and noise. Finally, they could run a population classification analysis to check how well each pair of texture family is

discriminated by the population activity in each area.

Incidentally, it is interesting to point out that, based on the results of Bolanos et al, 2024, in the mouse, it is area LM to carry out more info about textures, as compared to V1. And LM has been equated to monkey V2 by several anatomy studies (Gao et al., 2010; Wang et al., 2011, 2012). I believe this consistency among monkeys, mice and tree shrews should be mentioned somewhere in the manuscript.

MINOR ISSUES

1) Page 3, Lines 125-130 "... receptive field positions were concentrated in a small portion of the screen ... Surprisingly, clustering of receptive fields was apparent in all areas studied, even TI-ITi and ITr ... maintain retinotopic organization; again, this contrasts with the primate, where retinotopy is absent in anterior temporal cortex."

Two comments here:

A) The fact that "receptive field positions were concentrated in a small portion of the screen" and "clustering of receptive fields was apparent in all areas studied" does not mean anything per se, because the level of overlap between RF positions will depend on the amount of cortical surface spanned. And this, in turn, will depend on the inclination of the probe during the recording session. A probe inserted perpendicularly to the target area will not allow spanning much cortical surface and, because of the retinotopy, RFs will all tend to be close to each other. A probe inserted diagonally, with a large angle, will span instead a lot of cortical surface and therefore the RFs will span a large portion of the visual field. It is unclear here whether such concentration of RF centers on very similar locations is simply due to the penetration being very much orthogonal to the cortical surface. Please clarify.

B) Regarding this statement "even TI-ITi and ITr, though located at the anterior end of the tree shrew ventral stream, maintain retinotopic organization; again, this contrasts with the primate, where retinotopy is absent in anterior temporal cortex", two comments:

First, it is too strong to say that there is no retinotopy in IT. While early studies in IT, as the one cited by the authors, reported very large RFs, spanning tens of deg, at least 3 studies have found very wide variations in RF size in IT, including RFs with just a few deg of size. Please refer to: (Op De Beeck and Vogels, 2000; DiCarlo and Maunsell, 2003; Zoccolan et al., 2007). It is true that there is a bias for RFs to be located close to the fovea but there is variation of several deg in their location (see Fig 6 in Op De Beeck and Vogels, 2000). So, this statement should be corrected to better reflect the literature and toned down.

Yet, it is true that, if you look at the medians across areas, as well as at the example RFs in fig. 1g, the increase from V1 to deeper areas is minimal. So, this is a place where some interesting comparison can be drawn with what found along the rat ventral stream, where, instead, a very sharp increase of RF size was reported. Median RF size double from V1 to LL in (Tafazoli et al., 2017) and almost doubled from V1 to TO in (Vermaercke et al., 2014). Also along the mouse visual cortical hierarchy the mean RF size increases dramatically (Wang and Burkhalter, 2007; Siegle et al., 2019), although in the mouse it is hard to restrict this analysis to ventral regions only (given that, in this species, compared to the rat, the ventral stream is way less understood at the functional level). I think that this comparison also with rodent data should be mentioned, given that it may suggest a larger affinity between monkeys and rats/rodents than monkeys and screws, at least for some specific functional properties. I also think that it would be interesting to add these comparisons to Fig. 5g.

2) Analysis of orientation tuning (Fig. 2a-d). The trend found for the orientation tuning (larger in V1/V2 and smaller in deeper areas) is an interesting result, but I would recommend the authors to expand the interpretation of this finding through a deeper comparison with the monkey, rodent and modeling literature. In fact, the question of whether orientation tuning should increase or decrease along a ventral hierarchy does not have a so obvious answer. Some rodent studies have indeed assumed that a ventral stream, given the need to process fine details of the image features, should retain or even amplify tuning for orientation. See for instance (Marshel et al., 2011).

By contrast, we have shown in (Matteucci et al., 2019) and, later, in (Muratore et al., 2022), that, along an object processing hierarchy (e.g., a deep net like VGG-16,) orientation tuning and, more in general, information about orientation, does decrease in the deeper layers, after an initial increase across the very initial few layers. As indicated by the authors, this is indeed consistent with what found along the monkey ventral stream. But, rather than referring to the single study they have cited here, they could refer to the meta-analysis of the monkey literature we carried out in Matteucci et al, 2019, where we pooled together data about orientation tuning from about 15 studies in monkey V1, V4 and IT (see Table 1 and Fig. 7D). Finally, in our study we did find a sharp decrease in orientation tuning along the rat ventral stream, which is very consistent with the one found by the authors in the tree shrew. It would be interesting however, if the authors, in addition to compute their metric based on explained variance, computed also the more traditional OSI index. This would allow a more direct comparison with our meta-analysis of former monkey studies and our finding on the rat. Also, it would perhaps be interesting this comparison to Fig. 5g.

3) Page 5, line 222-223 “we summed the total explained variance for each area by each AlexNet layer across cells”. This should be described better. The text says “we summed” but this would imply that the metric increases as a function of the number of cells and, since a different number of cells was recorded in each area, this would be a problem. I would have expected the authors to average the explained variance across cells rather than simply sum or perhaps take the medians of the distributions of explained variance across the recorded populations. Clarify please.

4) Page 7, lines 339-340 “they were comparatively small ... relative to the large spatially-invariant receptive fields of primate anterior IT cells (mean = 24.5 deg)”. As mentioned already, this number comes from a very old paper and has not been confirmed by more recent investigations. The most systematic study of RF size in IT reports a means of about 10 deg (Op De Beeck and Vogels, 2000) . Other studies have shown RFs that are even smaller - just a few degs. This statement should be toned down. It is true that the increase of RF along the shrew ventral stream is surprisingly modest, compared to both monkey but also rats and mice. Still, the monkey RF size in IT should not be overestimated.

5) Fig. 5g. Given that object/shape processing has been studied way more systematically and carefully in rats than in mice, I find it odd that the table include mice but not rats. A row describing the findings on rats should be added. Under the column view invariance, the entry should report yes: LL and TO, based on (Vermaercke et al., 2014; Tafazoli et al., 2017). Also, the entry of the texture column should be corrected and report LM, based on (Bolaños et al., 2024).

REFERENCES

- Baldassi C, Alemi-Neissi A, Pagan M, DiCarlo JJ, Zecchina R, Zoccolan D (2013) Shape Similarity, Better than Semantic Membership, Accounts for the Structure of Visual Object Representations in a Population of Monkey Inferotemporal Neurons. *PLoS Comput Biol* 9:e1003167.
- Bolaños F, Orlandi JG, Aoki R, Jagadeesh AV, Gardner JL, Benucci A (2024) Efficient coding of natural images in the mouse visual cortex. *Nat Commun* 15:2466.
- DiCarlo JJ, Maunsell JH (2003) Anterior inferotemporal neurons of monkeys engaged in object recognition can be highly sensitive to object retinal position. *J Neurophysiol* 89:3264–78.
- Gao E, DeAngelis GC, Burkhalter A (2010) Parallel Input Channels to Mouse Primary Visual Cortex. *J Neurosci* 30:5912–5926.
- Marshel JH, Garrett ME, Nauhaus I, Callaway EM (2011) Functional Specialization of Seven Mouse Visual Cortical Areas. *Neuron* 72:1040–1054.
- Matteucci G, Marotti RB, Riggi M, Rosselli FB, Zoccolan D (2019) Nonlinear Processing of Shape Information in Rat Lateral Extrastriate Cortex. *J Neurosci* 39:1649–1670.
- Matteucci G, Zoccolan D (2020) Unsupervised experience with temporal continuity of the visual environment is causally involved in the development of V1 complex cells. *Sci Adv* 6:eaba3742.
- Muratore P, Tafazoli S, Piasini E, Laio A, Zoccolan D (2022) Prune and distill: similar reformatting of image information along rat visual cortex and deep neural networks In *Advances in Neural Information Processing Systems* 34 .
- Op De Beeck H, Vogels R (2000) Spatial sensitivity of macaque inferior temporal neurons. *J Comp Neurol* 426:505–18.
- Siegle JH et al. (2019) A survey of spiking activity reveals a functional hierarchy of mouse corticothalamic visual areas. *bioRxiv*:805010.
- Tafazoli S, Safaai H, De Franceschi G, Rosselli FB, Vanzella W, Riggi M, Buffolo F, Panzeri S, Zoccolan D (2017) Emergence of transformation-tolerant representations of visual objects in rat lateral extrastriate cortex. *eLife* 6:e22794.
- Vermaercke B, Gerich FJ, Ytebrouck E, Arckens L, Op de Beeck HP, Van den Bergh G (2014) Functional specialization in rat occipital and temporal visual cortex. *J Neurophysiol* 112:1963–1983.
- Wang Q, Burkhalter A (2007) Area map of mouse visual cortex. *J Comp Neurol* 502:339–357.
- Wang Q, Gao E, Burkhalter A (2011) Gateways of Ventral and Dorsal Streams in Mouse Visual Cortex. *J Neurosci* 31:1905–1918.
- Wang Q, Sporns O, Burkhalter A (2012) Network Analysis of Corticocortical Connections Reveals Ventral and Dorsal Processing Streams in Mouse Visual Cortex. *J Neurosci* 32:4386–4399.
- Zoccolan D, Kouh M, Poggio T, Dicarlo J (2007) Trade-off between object selectivity and tolerance in monkey inferotemporal cortex. *J Neurosci* 27:12292–307.

Version 1:

Reviewer comments:

Referee #1

(Remarks to the Author)

The new article is well revised, and the addition of new data from macaque to provide quantitative comparison is excellent. Figure 6g is a fundamental conclusion that will be often reprinted and examined. The authors have done a good job addressing all the concerns.

I have only a very small comment. In Figure 6g, there are no black lines on the primate side (face cell decoding). I don't think it is critical that they be added, but as a reader, I found myself not understanding why they weren't there. Some small note in the caption as to why they are present on the tree shrew side and not on the primate side would be very helpful. (If you can do the same analysis and add them that might be best, because I think that panel is going to be reproduced in many talks, reviews, textbooks, etc.)

**Referees' comments:**

We sincerely thank both reviewers for their thoughtful and constructive feedback. Their insights have greatly
improved the manuscript. We have carefully addressed each of the reviewers' concerns and provide
detailed responses to their points below.

**Summary of Major Changes**

Before addressing the individual comments of the reviewers, we would like to summarize the major new
experiments and analyses we performed since the last version:

1) We have added extensive additional controls to address the reviewers' concerns about the influence of
low-level features in our data. This includes controlling for luminance, contrast, and spatial frequency. We
took great care to make sure to control for this also at the local level (i.e. inside each cell's receptive field).
Overall, these new analyses demonstrate that our findings in tree shrew V2 cannot be accounted for by V2
simply being more sensitive to low-level features. We believe these analyses significantly strengthen our
findings and provide further evidence that V2 harbors complex representations for object processing.

2) To strengthen the comparative nature of our manuscript, we have added new recordings from macaque
area V2 and two regions from macaque IT. Thus, we provide newly collected data showing recordings from
homologous areas in two species using the same Neuropixels probes, the same 1593 images, and same
data analyses. Primate IT cortex has long remained the cornerstone for studying processing of complex
objects such as faces. The side-by-side comparison reveals that macaque V2 is fundamentally different
from tree shrew V2, and does not share the high-level coding properties of the latter. Our new data strongly
supports our original thesis, that the tree shrew visual system harbors a compressed hierarchy performing
many of the same functions as primate IT but earlier, at the level of area V2.

3) Finally, we also include additional changes to the text and figures to aid clarity based on the reviewer's
suggestions. In light of the comparative nature of our manuscript, we provide a more encompassing
comparison of our findings to that in rodents, especially the rat. We believe this will further add interest to
a broad readership and will hopefully foster more comparative studies of this nature.

**Referee #1 (Remarks to the Author):**

The manuscript by Lanfranchi and colleagues describes the results of a creative approach to comparative
analysis of visual systems across mammals. The authors used Neuropixels probes to examine visual
responses in many tree shrew brain areas to a large stimulus battery. The authors find that cells in area V2
exhibit selectivity for complex objects that is generally not found in area V2 of the primate. The analysis and
basic recordings are well done. The conclusion hinted at, that animals with fewer visual brain areas must
have processing networks that are less deep or differently constructed (more recurrency than feed-forward,
a possibility raised by the authors) is very interesting and would appeal to the broad readership of Nature.
The figures are clear and easy to interpret.

I have two major concerns that I believe the authors can address.

First, I wonder if the results follow in a trivial manner from the lower spatial frequency selectivity of the tree
shrew visual system as compared to the primate visual system. The stimuli being shown have high spatial
frequency characteristics, and clearly cannot be reconstructed as well they could be from recordings from
the primate visual system. If one were only going to reconstruct low spatial frequency information, would
the hierarchical nature of the primate visual system be less pronounced? This concern applies to the main
result of the paper (the early emergence of object-decoding responses already in V2) as well as to the
conclusions about view invariance in Extended Data Figure 4. I imagine that other readers may have the
same question, so it would be important to address this issue in the text and figures rather than just in a
response to the reviewers.

Second, the primate visual system is the comparative anchor of the narrative, yet comparable data are not
shown. I realize that the authors likely do not have data from many of these putatively homologous areas,
but do the authors at least have data from primate V2 and IT that can be used in even a rough comparison?
It would make the paper much stronger to have more than a by-narrative-only comparison (the data in
Expanded Data Figure 4 is only for IT) of what we expect V2 to do under the same stimuli.

With those criticisms mentioned, I think the paper is creative and brings a quantitative edge to comparative
studies across mammals that is paradigm-shifting and will appeal to the Nature readership.

We sincerely thank the reviewer for their comments and valuable suggestions. We will begin by addressing
the reviewer's second concern. Our response to their first concern follows after that (because data from the
monkey actually helps address this concern).

We completely agree that the comparison between tree shrew and monkey data is extraordinarily important.
Given that we are fortunate to have access to the newly developed NHP Neuropixels probes, we decided
to perform large-scale recordings in monkeys to provide a direct comparison to our tree shrew dataset
(**Reviewer Fig. 1a-c**). We presented the same 1593 object stimuli while recording from V2, posterior IT,
and anterior IT from two monkeys per area. These data are included in **Fig. 6**. To summarize our findings:

- • Strikingly, we found that accuracy for decoding object identity from primate V2 was far lower than
that from tree shrew V2 (**Reviewer Fig. 1d**). In fact, **tree shrew V2 was similar to that of primate**
**posterior IT**. As expected, primate anterior IT that sits at the apex of the primate ventral visual
stream outperformed all other areas.
- • We did not observe strongly face-selective cells in our recordings from primate V2, in contrast to
tree shrew V2 (**Reviewer Fig. 1e**).
- • Both primate V2 and primate posterior IT were best explained by middle layers (Conv5) of AlexNet,
while primate anterior IT was best explained by late layer FC7 (**Reviewer Fig. 1f**). In the tree shrew,
V2 was best explained by Conv5. Thus, this particular comparison is not diagnostic of whether tree
shrew V2 corresponds to macaque V2 or macaque posterior IT, but does suggest that macaque
anterior IT is a new evolutionary development.
- • Tree shrew V2 supported the most accurate object reconstruction among all tree shrew visual
areas, while in macaque anterior IT performed the best (**Reviewer Fig. 1g**).

Reviewer Figure 1. Comparison of responses in tree shrews and primates along the visual hierarchy. (Panel a, b, c, d, e, f, g replicated from Fig. 4 and 6):

(a) Schematic showing Neuropixels NHP 1.0 probes recording from V2, posterior IT, and anterior IT in macaques. **(b)** Representative coronal slices from each area targeted with a single electrode. **(c)** Responses of cells in each area to 96 stimuli composed of faces and objects (columns). Only visually responsive cells were included (t -test, $p < 0.05$). **(d)** Decoding performance for individual object identity (dashed lines) or face identity (solid lines) as a function of number of cells used by the classifier for tree areas (top) and macaque (bottom) areas. **(e)** Histograms of t -scores for face selectivity across areas. **(f)** Normalized explained variance by AlexNet layers for each visual area. **(g)** Average decoding distance for each visual area between AlexNet FC6 activations predicted from neural activity and actual FC6 activations for each image, normalized by theoretical best decoding distance (*Methods*).

Together these data support our findings that the tree shrew contains a compressed hierarchy, with tree
shrew V2 performing functions akin to primate IT, and importantly, *primate V2 does not show these*
*characteristics*. They further clarify that tree shrew V2 is specifically most analogous to primate posterior
IT.

We now clarify these points in the Discussion:

“As a direct comparison, we performed the same experiments across three homologous regions in the macaque, including
V2, IT anterior and IT posterior. Strikingly, this comparison further supports our finding that the tree shrew contains a
compressed hierarchy that performs many of the functions akin to primate IT in tree shrew V2, and importantly, primate V2
does not show these characteristics. Our stimulus set was originally tailored for primate object recognition and, therefore,
facilitated direct comparisons to primates. However, future work is needed to expand these findings with additional stimulus
sets such as those that include ethologically relevant objects, view invariance, binocular disparity and multimodal stimuli. In
fact, studies in rats using highly controlled visual morphed objects designed to match luminosity across transformations led
to the discovery of properties of object processing in the rat visual area LL and TO including view invariance (Vermaercke
et al., 2014; Tafazoli et al., 2017). Such comparisons in the tree shrew would further shed light into evolutionarily conserved
properties for object vision that places this species functionally between the primates and rodents.”

Regarding the reviewer’s first point that the early emergence of object processing in tree shrew V2 may in
part be accounted for by the limited ability of tree shrews to resolve high spatial frequencies, we performed
several new controls to address this concern. First, we found that image decoding using responses from
tree shrew V2 was actually comparable to that from monkey posterior IT and better than that from monkey
V2 (**Reviewer Fig. 1d**). This result is notable because 1) it confirms our prediction that tree shrew V2
already harbors high level computation previously found only in primate IT; and 2) it confirms that lower
visual areas in the primate--unlike the tree shrew--do not perform well at image decoding. This suggests
that the high decoding performance of tree shrew V2 is not simply due to the tree shrew only seeing low
spatial frequencies; if this were the case, macaque V2 should have shown decoding performance
comparable to tree shrew V2.

Second, to directly assess whether spatial frequency filtering may explain differences in hierarchical
organization between tree shrew and monkey, we divided our images into seven groups based on the
relative preponderance of high vs. low frequencies in these images (**Reviewer Fig. 2a, b**). We then
measured what percentage of firing rate variance in the various brain areas was explained by image identity
within each image group (**Reviewer Fig. 2c, d**). While we found minor differences between the seven
groups, these differences were much smaller than the differences between areas: For all seven groups of
images, the percentage of explained variance in tree shrew V2 was much higher than in tree shrew IT. In
contrast, in the monkey, the fraction explained variance was higher in anterior IT than in V2 for all seven
groups. Thus, differences in ability to represent high-frequency image features appear unlikely to explain
differences between tree shrew and monkey hierarchies.

[FIGURE REDACTED]

b

c

d

Reviewer Figure 2. Spatial frequency characteristics do not account for the early emergence in object processing in the tree shrew. (Panel a, b, c replicated from Extended Data Fig. 2f, g, h):

(a) [TEXT REDACTED] (b) Power spectrum across groups of images in (a) relative to the middle spatial frequency group. (c) Percentage of variance of neural responses explained by object stimulus identity in each area, separated into categories based on spatial frequency. (d) Same as (c) but for macaque areas.

**Small items:**

**Line 70: reword, sounds like the primates are moving instead of the reader's attention**

We changed the sentence as follows:

"We confirmed hallmarks of hierarchical organization found in primates including increased receptive field size and response
latency (Siegle et al., 2021) as well as increased selectivity for naturalistic textures compared to spectrally matched noise
(Freeman et al., 2017) moving anteriorly along the tree shrew visual pathway."

**Line 111: "responded to any of the classes of visual stimuli we tested". There must be some statistical
definition of "responded" yes? Please indicate.**

Yes, this is true. We have now added a note to refer to the methods section to find the statistical definition.
The definition is as follows in the methods: "Visually responsive cells: A cell was deemed responsive to a
particular class of stimuli (either gratings, textures, and noise, or faces and objects) if its average firing rate
in the 100 ms following stimuli of that class exceeded the expectation value based on a Poisson model
trained on the firing rate in the 50 ms before all the stimuli of that class."

**Line 303: "A hallmark of the ventral stream" -> "Another hallmark" (phrase was already used in line 258)**

We removed the repeated phrase and changed the sentence as follows:

"The primate IT cortex is organized into subregions containing cells that respond maximally to images from specific
categories, e.g., faces."

**Line 611: is there a citation that can be provided for the explainable variance?**

We have now added a citation for the explainable variance (Bao et al., 2021).

**Line 639: the paragraph is a little brief, please expand**

This method section refers to object reconstructions. We have now provided references to previous papers
from our lab that have extensive information in the methods sections. In addition, we have expanded this
part of the methods section for clarity as follows:

Object reconstruction and normalized decoding distance

Image reconstructions were performed as previously described in (Bao et al., 2021) and (Vadia et al. 2024). To generate
images that reflect the features encoded in the neural responses, we passed into AlexNet images from an auxiliary database
comprising a much larger set of 15901 images, none of which was previously shown to the animal. For each stimulus image
presented to the animal, the feature vector decoded from the neural activity was compared to the feature vectors of the
larger auxiliary stimulus set. We defined the "reconstructed image" as the image in the auxiliary dataset with the smallest
Euclidean distance to the decoded feature vector of the original image.

Given that the auxiliary images used for reconstruction did not include any of the objects shown to the animals (limiting how
good the reconstruction can be), we computed a 'normalized decoding distance' to quantify the reconstruction accuracy for
each object. We first used the Moore–Penrose pseudoinverse to transform the predicted features from neuronal data back
into the space of AlexNet layer FC6 activations. Next, we calculated the Euclidean distance between these pseudoinverted
predicted features and the actual AlexNet FC6 activations deriving from the presented images. We normalized this distance
by the theoretical best decoding distance, i.e. the distance between the actual AlexNet FC6 activation and the back
projection of the 50D PCA output of AlexNet FC6 (again using the Moore–Penrose pseudoinverse). Thus, the normalized
decoding distance for an image is:

$$\text{Normalized decoding distance} = \frac{|V_{\text{recon}} - V_{\text{original}}|}{|V_{\text{best possible recon}} - V_{\text{original}}|}$$

where $\mathbf{v}_{\text{recon}}$ is the feature vector reconstructed from neuronal responses, $\mathbf{v}_{\text{original}}$ is the feature vector of the image presented
to the animal, and $\mathbf{v}_{\text{best possible recon}}$ is the feature vector of the best possible reconstruction. A normalized distance of one
means that the reconstruction has found the best solution possible.

**Extended Figure 1b: What is the counterstain of the tissue? Please include its name in caption and
procedures in methods.**

The counterstain is DAPI, and it is now included in the methods and figure legend.

Discussion: One idea the authors may consider adding to the discussion is the possibility that the tree
shrew's visual system may be configured to process additional stimuli that were not in the stimulus set used
by the authors. For example, if the authors had used stimuli with more temporal structure or binocular
disparity, they might have observed a more hierarchical organization of optimal reconstruction. The paper
reads as though the authors believe the chosen stimulus set can reveal all we need to know about object
recognition, but this is really an assumption. It is clearly an interesting stimulus set to examine, but it might
be really tailored to primate object recognition (something we can relate to as humans and well represented
by IT) rather a general representation of stimuli important to all mammals. I don't think this issue detracts
from the paper or need to alter the primary narrative, but I think noting it would add depth to the space of
interpretations.

We completely agree, and have now added the following statement to the Discussion:

"Our stimulus set was originally tailored for primate object recognition and, therefore, facilitated direct comparisons to
primates. However, future work is needed to expand these findings with additional stimulus sets such as those that include
ethologically relevant objects, view invariance, binocular disparity and multimodal stimuli."

Anecdotally, we note that we took various real laboratory objects (e.g., screwdriver, computer mouse, etc.)
and waved them in front of the tree shrew to see if these would elicit stronger responses in V1, V2, *Pulv*,
TP, TI-ITi and ITr, than our set of 1593 images. We never found that this made a big difference.

**Referee #2 (Remarks to the Author):**

In this manuscript, the authors explore whether the anatomical hierarchy of visual areas that, in the tree
shrew, is the homologous of the macaque ventral stream harbors an increase of complexity of tuning and
encoding properties, with respect to the image space, that would make it homologous to the monkey ventral
stream also at the functional level. The authors record the activity from hundreds of neurons in 6 different
visual areas of the tree shrew, probing their responses with a rich battery of stimulus conditions, including
gratings, textures and objects. They find that some tuning properties follow a monotonic trend along the
anatomical hierarchy (e.g., orientation tuning decreases), while other properties (such as tuning for visual
objects, ability to discriminate face identity, etc) peak already in V2, rather than in the deepest regions of
the pathway. Hence the authors' conclusion of a compressed processing hierarchy in this species.

I found the scientific question very interesting and the study well crafted, although I believe it has two major
weaknesses. At the data analysis level, the authors have not explored the extent to which the trends they
found can be explained by lower-level areas (V1 and V2) being more sensitivity to low/middle level features
(such as luminance, contrast, aspect ratio, surface area of the objects). At the experimental design level,
they did not include in their stimulus sets any conditions to probe the invariance of the representations,
which, among all properties that can be studied along a ventral-like stream, is not only the most relevant
but also the one that can be measured more rigorously. Hard to make definite claims about whether an
anatomical progression behaves as an object-processing pathway unless invariance is systematically
assessed.

We are grateful to the reviewer for his constructive feedback and suggestions, which have greatly helped
232 us to improve the clarity and strengthen the presentation of our findings. We are especially grateful for the
233 detailed analysis suggestions motivated by the reviewer's previous work which helped us greatly to clarify
our conclusions. In particular, we understand the importance of controlling for low-level features in our data
(a point also raised by reviewer 1) and have now provided the suggested essential controls. Overall, these
new analyses demonstrate that our findings in V2 cannot be accounted for by V2 simply being more
sensitive to low-level features.

We unfortunately no longer have tree shrews in the laboratory and therefore are unable to collect further
data to explore invariance. However, we have performed an additional experiment recording from macaque
V2 to validate our previous predicted view invariance analysis. We agree that the previous analysis lacked
this important control. We have also removed any strong claims about the lack of invariance in the tree
shrew. This will be an important avenue for further studies on object-processing pathways in the tree shrew.
We now state in our discussion:

"As a direct comparison, we performed the same experiments across three homologous regions in the macaque, including
V2, IT anterior and IT posterior. Strikingly, this comparison further supports our finding that the tree shrew contains a
compressed hierarchy that performs many of the functions akin to primate IT in tree shrew V2, and importantly, primate V2
does not show these characteristics. Our stimulus set was originally tailored for primate object recognition and, therefore,
facilitated direct comparisons to primates. However, future work is needed to expand these findings with additional stimulus
sets such as those that include ethologically relevant objects, view invariance, binocular disparity and multimodal stimuli. In
fact, studies in rats using highly controlled visual morphed objects designed to match luminosity across transformations led
to the discovery of properties of object processing in the rat visual area LL and TO including view invariance (Vermaercke
et al., 2014; Tafazoli et al., 2017). Such comparisons in the tree shrew would further shed light into evolutionarily conserved
properties for object vision that places this species functionally between the primates and rodents."

I can say this by direct experience, given that I embarked in a journey that is very similar to the one of the
authors, having tried to establish whether the rat visual cortex contains a functional object processing
hierarchy akin the ventral stream. Full disclosure here: I am Davide Zoccolan, the author of several studies
on the rat ventral stream that addressed very similar questions to those addressed here by the authors. I
prefer to disclose my name, since many of the criticisms and suggestions I have listed below derive from
my very direct experience in dealing with similar data and questions. So, I believe that all my comments will
be more understandable if the authors know where they come from.

Connected to this, another general criticism I have for the authors is that they have not made any attempt
to compare their findings with any species other than monkeys. Rodent studies are almost fully ignored.
This is unfortunate given the evolutionary position of tree shrews (somewhat in between rodents and

monkeys) and the efforts that have been made to understand whether rodents possess functional
processing hierarchies similar to those of primates.

We agree and apologize for not adequately citing the previous literature on rodents. In fact, we were inspired
by work in rats at the start of this study to investigate object processing in the tree shrew. We are thankful
for these suggestions and now include more comparison to the rodent literature in our introduction and
discussion. For comparative purposes, we have expanded our schematic in **Fig. 6h** to include mice, rats,
tree shrews and macaque.

**MAJOR ISSUES**

1) Page 4, Line 195-197 “After V2, the explainable variance in responses to these complex object stimuli
decreased dramatically ... This contrasts with the primate, where, e.g., ~80% of variance of IT cell
responses to a large image set can be explained by image identity. Overall, these results suggest that even
though anterior tree shrew areas were relatively more responsive to complex objects than to gratings, their
responses to these complex object stimuli were not highly reproducible from trial-to-trial”. This result has
not to do only with the reproducibility of the response but also with how similar (in units of variance) the
mean responses to each stimulus are. So, the conclusion is that the discriminability of individual object
conditions decreases from V2 onward. But this finding, without further analysis, does not mean, per se, that
deeper temporal areas are less good at encoding high-level visual information. Because of two reasons.

First, when stimuli as complex as objects are presented, there is a very large number of low-to-middle level
features that may systematically vary across the objects. For instance, the luminance and contrast that
impinge upon the RF of each unit. In our previous study of the rat ventral stream (Tafazoli et al., 2017), we
quantified these properties and referred to them as effective RF luminance and contrast. In that paper, we
carried out an analysis similar to the one of the authors, but based on the mutual information $I(R;S)$ between
stimulus identity (S) and response (R) and we found a very peculiar trend. $I(R;S)$ decreased sharply along
the ventral stream (being maximal in V1), similarly to what reported here by the authors. But we discovered
that the reason was that units in earlier areas carried way more info about low level features of the objects
(luminance and contrast). In other words, V1 units discriminated better among object conditions simply
because they were more sensitive to the fact that some objects were brighter or larger or (conversely)
darker or thinner. By comparison, LL, the deepest rat ventral area, carried way less info about such low-
level properties. We found that this was the reason at the root of its lower $I(R;S)$. But, once we separated
the part of the mutual info about low level properties (e.g., luminance; white bars in Fig. 3B of Tafazoli et al)
from that of higher-level features (colored bars in Fig. 3B), it turned out that the latter was equally large in
all areas. Therefore, LL carried the largest percentage of higher order info about stimulus identity than all
other areas (Fig. 3C). Interestingly, we have shown that this same behavior can be found for the mutual
info conveyed by the units of deep nets (VGG family) about luminance and contrast (Muratore et al., 2022).

Looking at the examples of stimuli used by the authors, it seems that they vary substantially in terms of
object area, contrast, luminance etc. So, their stimulus set lends itself to the same possible confound of our
stimulus set, given the analysis they applied. In other words, the decrease of explained variance across the
shrew ventral stream could possibly reflect a decrease of sensitivity to low level properties. But pruning
such info would be a key functional feature of a ventral pathway. So, it is essential that the authors control
for that. Formally, the best way to do it would be to compute the mutual info between stimulus and response
and then break it down in the low-level and higher-level components, as we did in our study, by computing
conditional mutual info. The methods section of our paper explains in details how this can be done, but
dealing with mutual info analysis applied to neuronal data is always technically tricky. However, the authors
could perhaps look at simpler ways to do this analysis: for instance, compute the amount of variance
explained by luminance and contrast and then by aspect ratio of the objects or surface area. It would
perhaps be enough to regress the responses to these continuous variables and compute the R2. Care
should be taken however to quantify how much contrast and luminance entered the RF of each specific
unit (again this is described in the method section of our paper). Without such control, it would be unreliable
to conclude that there is not a hierarchical increase of object information along the ventral stream of the
tree shrew.

We agree that mutual information (MI) is a very attractive formalism for this kind of analysis. We followed
the formulas in Tafazoli et al., 2017 and obtained the MI between the presented images and neuronal activity
(see the box whisker plot below representing the probability density function across cells).

Reviewer Figure 3. Mutual information for low-level object features:

(a) Mutual information (MI) between the presented images and neuronal activity. (b) Conditional mutual information of luminance within cells' RF. Note different vertical scale.

We observed that V1, V2 and ITr were significantly different from other areas in terms of the MI of their neurons' firing rates with image identity (**Reviewer Fig. 3a**). By replicating the procedure in Tafazoli et al., 2017, we then proceeded to calculate the conditional mutual information of the firing rates with the luminance of the object images within the cell's RF (**Reviewer Fig. 3b**). We found tree shrew neurons carried very little information about this low-level feature of the objects, even in early areas V1 and V2.

However, as the reviewer points out, we acknowledge that a fundamental difficulty with applying the MI formalism to our data is the fact that we have a very large set of different stimuli (1593) and at most 10 repeated presentations of each. As a consequence, the probability space of neuronal responses (number of spikes per presentation) is sparsely sampled. This leads to relatively poor estimates for the joint probabilities that appear in the equation for MI, which in turn leads to poorly estimated MI for each cell. In addition, the natural way to combine MI across cells involves calculating the MI between the images and the joint probability of firing rates across cells. This, however, vastly exacerbates the problem of estimating pdfs, because the joint pdf of firing rates is exponentially more sparsely sampled still. Accordingly, we also followed the reviewer's other suggestion and determined the fraction of variance that could be explained by various low-level features.

To do so, we again followed the same methodology from Tafazoli et al., 2017 and calculated for each individual neuron how much contrast and luminance entered its RF for each of the images. We found that luminance and contrast only explained a small fraction of the variance of the neuron's image responses (**Reviewer Fig. 4a-c**). Thus, even after accounting for tuning to local luminance and contrast, area V2 still showed the highest explained variance by image identity. We agree that controlling for low-level features was crucial and we therefore added this analysis (see below) to **Fig. 3c** and **Extended Data Fig. 2b**.

[FIGURE REDACTED]

Reviewer Figure 4. Population variance explained by luminance and contrast (Panel a, b, c replicated from Extended Data Fig. 2c, d, e):

(a) [TEXT REDACTED] (b) Percentage of variance of neural responses explained by object stimulus identity in each area. Dark bars correspond to the part of the variance accounted for by luminance impinging each receptive field. (c) Same, but dark bars correspond to contrast.

In addition to local luminance and contrast, we also explored a diverse selection of other low-level image
features. Specifically, we calculated feature indices (area, circularity, compactness, contrast, eccentricity,
edginess, horizontality, internal contrast, luminance, orientation, power ratio, solidity and stripiness) for each
image and asked to what extent these indices explained the neural responses in each area. We used linear
regression to calculate the fraction of variance of each cell's responses explained by each feature
(Reviewer Fig. 5a, b). We found that none of these indices explained more than around one percent of the
variance in our data. Furthermore, we found that the amount of explained variance by image identity that
could be accounted for by *any* of the indices was less than half, and V2 remained the area with highest
explained variance even after subtracting the contribution of these low-level features (Reviewer Fig. 5c).
Overall, these results show that the reviewer is absolutely correct, and low-level features do explain a
portion of the variance in each area, but they cannot account for the superiority of V2 over more anterior
areas in the amount of variance that can be explained by image identity.

[FIGURE REDACTED]

Reviewer Figure 5. Indices for low-level image features account for only a small fraction of variance explained. (Panel a and b replicated from Extended Data Fig. 2a, b. Panel c replicated from Fig. 3c): (a) [TEXT REDACTED] (b) Histogram indicating the average fraction of variance in the firing rate explained by various low-level image feature indices. (c) Percentage of variance of neural responses explained by object stimulus identity in each area (left bars) and the amount of variance that can be explained by low-level feature image indices (right bars).

2) Page 5, lines 224-228 "... early visual areas V1 and V2 were best explained by early layers Conv3 to
Conv5, whereas TI-ITi and ITr were best explained by FC6 (Fig. 4c). However, the absolute explanatory
power of AlexNet was lower for the higher cortical areas ...". The comparison between relative and absolute
fraction of variance explained by Alexnet layers in each area is very interesting. One could interpret these
findings in different ways, but one possibility could be the following. Tuning in higher areas result from a
larger number of nonlinear hierarchal transformations and, as such, is much harder to model, even with the
latest stages of a deep net. In addition, in these areas, a big chunk of low-level visual info may have been
lost, which is the easiest to model. Hence, again, the difficulty of capturing a lot of variance (in absolute
terms) in the responses of neurons in the highest visual areas. Yet, the deepest Alexnet layers are the best
at modeling these neurons, which means that, in relative terms, a deep functional hierarchy is required to
capture some of the tuning of these cells.

Conversely, in V1/V2, responses may be dominated by a strong sensitivity to low level features, less
nonlinear transformations have accumulated, and, therefore, overall, regardless of the Alexnet layer used
to regress the neuronal data, the absolute amount of explained variance will be higher than in deeper visual
cortical areas. Yet, in relative terms, the larger amount of variance will be explained by early-middle layers,
given that the complexity of shape representations in V1/V2 is not that large.

In other words, these results are not necessarily in contrast with the existence of a functional hierarchy
where visual features' combinations of increasing complexity are progressively encoded. I believe the
authors should consider and discuss this scenario and try to test it, ideally by measuring how much tuning
for low-level features is present in the different areas.

We completely agree with the reviewer's main point here that a functional hierarchy may exist, and AlexNet
may simply not be expressive enough. Indeed, downstream areas such as TI and IT are suggested to be
multimodal (Wong and Kaas, 2009) which would arguably put them at a higher level of hierarchy than purely
visual areas. We now state:

"This analysis revealed early visual areas V1 and V2 were best explained by early layers Conv3 to Conv5, whereas TI-ITi
and ITr were best explained by FC6 (Fig. 4c). However, the absolute explanatory power of AlexNet was lower for the higher
cortical areas (Extended Data Fig. 2a, b), consistent with the lower explained variance by image identity in anterior areas
(cf. Fig. 3c). This could suggest that AlexNet may not be expressive enough to capture the response properties of tree
shrew IT as they have been suggested to be multimodal (Wong and Kaas, 2009)."

However, within the realm of vision, as we have already argued above (Reviewer Figs. 3-5), our analyses
indicate that tuning to low-level features cannot account for the unusually high explained variance by image
identity that we observed in tree shrew area V2.

Furthermore, we have now recorded in V2, posterior IT, and anterior IT in the monkey. Using the exact
 same analysis methods as in the tree shrew, we find a very different pattern of results in the macaque,
 where the highest explained variance by image identity occurs in anterior IT, not in V2 (**Reviewer Fig. 2c,**
 **d**); similarly, decoding performance steadily increases from V2 to posterior IT to anterior IT. This suggests
 that at least in the monkey, tuning in higher areas is not harder to model by AlexNet than tuning in lower
 areas (**Reviewer Fig. 1f**). Tree shrew V2 shows comparable decoding performance to macaque posterior
 IT (**Reviewer Fig. 1d**). Thus overall, unless we assume that the tree shrew possesses even more nonlinear
 hierarchal transformations than the macaque, the decreased explained variance by AlexNet features in tree
 shrew areas anterior to V2 cannot be accounted for by the inadequacy of AlexNet for modeling nonlinear
 visual features.

 3) Page 5, lines 232-233 “earlier PCs explained more variance in neural responses, with some variability
 across areas”. Related to my two previous points: this finding could suggest that earlier PCs of the Alexnet
 representation could be related to lower/middle level visual properties (local luminance, contrast, aspect
 ratio, surface area of the objects) and that visual neurons are very sensitive to such low-level attributes.
 The authors could perhaps measure these low-level properties across their stimulus set and see how well
 they correlate with the first few PCs of the Alexnet representation.

We measured several low-level properties, including the ones mentioned by the reviewer, for all images
 across the stimulus set and looked at how well they correlated with the first 10 PCs of AlexNet FC6
 (**Reviewer Fig. 6a**). Some of the low-level features do indeed show a higher correlation with the first two
 PCs than with later PCs. However, if we compare the explained variance by these low-level features with
 their correlation to PC1, we find there is no relation (**Reviewer Fig. 6b**). We emphasize that we are certainly
 not claiming that V2 shows no tuning low-level features; as previous analyses already establish that it does
 (**Reviewer Fig. 4c, d, Reviewer Fig. 5a, b, c**).

 Thus, although some of the low-level properties do mildly correlate with FC6 PCs, overall, these correlations
 were small. Importantly, we found that the degree of correlation for an individual feature poorly predicted
 the explanatory power of that feature over V2 activity.

**Reviewer Figure 6. Correlation of indices with low PCs poorly predict explanatory power of indices over V2 activity:**
 (a) Absolute correlation between image indexes and the first 10 PCs of AlexNet FC6. (b) Scatter plot between the percentage of explained variance by a given feature in V2 neural responses and the absolute correlation of that feature with FC6 PC1 activations in AlexNet.

 4) Page 5/6, lines 242-256 “...we asked whether activity in V2 might be sufficient to reconstruct objects
 using small neural populations ...”. I was not able to understand this analysis and, therefore, I cannot
 comment on the results of the image reconstruction. I urge the authors to expand considerably the Method
 section that describes the procedure, ideally also adding a supp figure to guide the reader.

We apologize for lack of clarity. As the reviewer suggested, we have added an extended figure to guide the
reader with schematics for how the method works (**Extended Data Fig. 3d**). We also have updated the
methods section and provided references to previous work from our lab with extensive methods details for
object reconstructions. The methods section is as follows:

Object reconstruction and normalized decoding distance.

Image reconstructions were performed as previously described in (Bao et al., 2021) and (Vadia et al. 2024). To generate
images that reflect the features encoded in the neural responses, we passed into AlexNet images from an auxiliary database
comprising a much larger set of 15901 images, none of which was previously shown to the animal. For each stimulus image
presented to the animal, the feature vector decoded from the neural activity was compared to the feature vectors of the
larger auxiliary stimulus set. We defined the “reconstructed image” as the image in the auxiliary dataset with the smallest
Euclidean distance to the decoded feature vector of the original image.

Given that the auxiliary images used for reconstruction did not include any of the objects shown to the animals (limiting how
good the reconstruction can be), we computed a ‘normalized decoding distance’ to quantify the reconstruction accuracy for
each object. We first used the Moore–Penrose pseudoinverse to transform the predicted features from neuronal data back
into the space of AlexNet layer FC6 activations. Next, we calculated the Euclidean distance between these pseudoinverted
predicted features and the actual AlexNet FC6 activations deriving from the presented images. We normalized this distance
by the theoretical best decoding distance, i.e. the distance between the actual AlexNet FC6 activation and the back
projection of the 50D PCA output of AlexNet FC6 (again using the Moore–Penrose pseudoinverse). Thus, the normalized
decoding distance for an image is:

$$\text{Normalized decoding distance} = \frac{|V_{\text{recon}} - V_{\text{original}}|}{|V_{\text{best possible recon}} - V_{\text{original}}|}$$

where $\mathbf{v}_{\text{recon}}$ is the feature vector reconstructed from neuronal responses, $\mathbf{v}_{\text{original}}$ is the feature vector of the image presented
to the animal, and $\mathbf{v}_{\text{best possible recon}}$ is the feature vector of the best possible reconstruction. A normalized distance of one
means that the reconstruction has found the best solution possible.

5) Page 6, lines 258-301, i.e., Analysis of Fig. 5. Also: page 7, lines 343-346. Again, unless some control is
done about the possible encoding of low level features by V2, I am not convinced about the conclusions of
the authors concerning this analysis. This 2D projection on the 2 first PC of the representation in Alexnet
FC6 sounds very much to me like a projection on a subspace encoding quite low-level features. Although
the authors do not mention this, it is sort of implicit in the names they gave to the axes of the space: stubby
vs. spiky and animate vs. inanimate (which I would more appropriately call squared vs. roundish perhaps)
are properties that correlates strongly (e.g.) with object surface area and, therefore, aspect ratio and
luminance. This subspace reminds me of a result we found in monkey AIT (Baldassi et al., 2013), where
the first PC that best explained the selectivity of the recorded neurons was a very low-level property: object
surface area (basically, the stubby vs. spiky axis of the authors). In that study, we did take this as a sign of
the (surprising) fact that IT neurons, despite being so high level, still carry substantial info about stimulus
properties that are quite basic and low-level, as lower-level areas are supposed to do. But this is of course
not the whole story - because what IT should do (as shown in many studies by DiCarlo lab for instance) is
to encode also higher order features (beyond these first two PCs of Alexnet fc6), which lower areas do not
encode. In this sense, in my opinion, the fact that neurons in V2 spanned all the quadrants in the 2 PC
space is not necessarily a sign of a higher level representation, as compared to deeper areas. Actually, I
would expect that tuning in deeper areas is accounted less well by these first 2 PC - that neurons occupy
less territory in the 2 PC space (exactly because those neurons now encode features that go beyond the
two basic axes stubby vs. spiky or animate vs. inanimate).

We think the coverage of the entire object space defined by the first two PCs of AlexNet FC6 by the preferred
axes of tree shrew V2 cells is interesting and worth pointing out given our lab’s previous discovery that
primate IT is topographically organized according to the space spanned by these two PCs (Bao et al.,
2021). Furthermore, FC6 is a relatively deep, fully connected layer of AlexNet, and hence we think the
space spanned by FC6 is meaningful reflection of the principle features underlying high-level object
representation. We have now collected additional data from primate V2, posterior IT, and anterior IT.
**Reviewer Fig. 8** shows the distribution of the lengths of the projections of the preferred axes of neurons
onto the object space across each tree shrew and macaque area (**Reviewer Fig. 8**). Tree shrew V2 and
primate anterior IT most clearly evenly span the four quadrants. This was quantified by computing the mean
of the Rayleigh distribution fit to the data, which was most evenly spread in tree shrew V2 and macaque
anterior IT.

Reviewer Figure 8. Quantification of the spanning of object space along the four quadrants:

(a) Distribution of the lengths of preferred axis projections onto the 2D object space spanned by the first two PCs of AlexNet FC6, separately for neurons that have their preferred axes projecting onto each of the four quadrants (Q) of object space (I Q. blue, II Q. yellow, III Q. green, IV Q. red). Inset shows Rayleigh fit results for each quadrant, with bar heights representing the mean preferred axis length for each quadrant. (b) Same, for macaque areas V2, IT post and

In V2, there could be a very good representation of faces, simply because what V2 neurons encode are
 low level attributes that faces happen to all share: similar aspect ratio, object surface size, contrast and
 luminance. One way to demonstrate that there is a real encoding of faces would be to play with those faces
 and transform them in various ways, so as to test invariance for face encoding. This is exactly the issue we
 encountered in analyzing the rat ventral stream (Tafazoli et al., 2017). Rat V1 neurons, given their sensitivity
 to low-level attributes and given that objects of the same category (even despite of transformations) shared
 low level features, were the best at encoding object identity, even across view changes. Only after carefully
 matching the mean luminance (across transformations) between the objects to be decoded, we found the
 signature of a hierarchical increase in the ability of visual areas to encode object identity in an invariant
 way. Critically, a demonstration of this kind is currently missing from this study. One way to achieve that
 would be to run a face categorization analysis in which faces are pitched against objects with aspect ratio,
 surface area and luminance as similar as possible to those of the face class. I believe an analysis of this
 kind is necessary.

We agree with these comments, and in fact did find that many cells that strongly responded to faces also
 respond to other objects, some of which have face-like attributes. As such, we certainly believe that these
 responses may to some degree be driven by low-level attributes of faces. However, while this line of
 argument may explain how cells attain the ability to *classify* images as face vs nonface (i.e., decode image
 class; Fig. 5e), it does not explain how cells attain the ability to *individuate* face images (i.e., decode image
 identity; Fig. 6g).

We do not believe the tree shrew brain is specialized to discriminate faces per se. That is, we agree that
 there likely isn't a "real" encoding of faces in the way the reviewer asks us to prove. Presumably, V2 would
 be quite good at individuating images that are "like" the face images in low-level properties. Nonetheless,
 the fact that V2 is specifically good at picking the identity of a specific object or face as compared to all the

other areas is surprising and impressive. This suggests that V2 is performing important computations
relating to processing objects, which was unexpected.

Most importantly, since our original submission, we have recorded in V2 in the primate brain while
presenting the exact same image set. Applying the same analysis used on the tree shrew data revealed
hardly any cells in primate V2 with high t-scores for face selectivity (**Reviewer Fig. 1e**). Unsurprisingly, the
same analysis on anterior IT in the primate revealed many such cells. The fact that our results from shrew
V2 are in some respects more like primate IT than like primate V2 makes it unlikely that our results in tree
shrew area V2 are due only to low-level image properties.

6) Last paragraph of the Results: invariance analysis. I am sorry, but the authors cannot claim that they
tested the invariance of the representations across the shrew visual areas. As they acknowledged, their
stimulus set did not contain conditions (i.e., multiple views of the same objects) to perform this test. What
they have done here (resorting to modeling with Alexnet) is not acceptable as a test of invariance. One
cannot simply use the linear regression model with Alexnet units to simulate how the actual neurons would
have generalized their responses across transformations. There is no guarantee that the model will be so
faithful to allow measuring invariance without recording actual neuronal responses. Every article that has
measured invariance in the monkey and, more recently, in the rat ventral stream, has made sure to have
carefully designed conditions to do so.

I believe that the authors should not claim they have tested invariance and remove reference to this test in
their Results and Discussion. Unless they can add a new set of recordings, even perhaps from just a few
checkpoint areas along the hierarchy, to really measure invariance. If they have the possibility to do so,
they should pay, again, a lot of attention to possible confounds yielded by sensitivity to low-level visual
properties. In our study on the rat ventral stream (Tafazoli et al., 2017), the lowest areas (V1 and LM) were
actually the more view invariant, unless the decoding analysis was restricted to pairs of visual objects that
were well matched in terms of luminosity across transformations.

We agree that invariance to viewing conditions is an extremely interesting feature of visual processing
systems. However, as mentioned above, we no longer had tree shrews in the lab by the time we finished
writing this paper (following our move to UC Berkeley). As such, we unfortunately cannot perform additional
experiments that would test for invariance by presenting stimuli comprising multiple views of the same
objects. We understand the reviewer's reservations about using our AlexNet-based view invariance
prediction analysis and agree 100% that it is much less conclusive than directly measuring the view
invariance of tree shrew visual areas, and this should absolutely be thoroughly tested in future studies.

However, we would like to point out that view invariance is not the only property defining a visual hierarchy.
We believe that the many other dimensions along which we searched for hierarchy (receptive field size,
latency, orientation tuning, spatial frequency tuning, texture tuning, object decoding, and reconstruction
capability) are all informative, and they all generally came to the same conclusion, that area V2 contains
the most sophisticated visual representation across all the areas, and later areas did not show clear
evidence for harboring a high-level representation (e.g., receptive field sizes were not significantly larger,
unlike in primate IT and rat LL and TO). Indeed, work in other species (Siegle et al., Nature 2021) has also
argued for the existence of hierarchical organization without a direct test of invariance.

Furthermore, in defense of our predicted invariance index approach, *we have now recorded from macaque*
*area V2, posterior IT, and anterior IT*. When we apply our predicted invariance index approach to this data
set, we find that the predicted invariance indices consistently overestimate view invariance compared to
ground truth, but nevertheless, *there is a positive correlation between the predicted and actual invariance*
*indices* (V2: $r = 0.166$, $p < 0.021$; IT post: $r = 0.189$, $p < 10^{-5}$; IT ant: $r = 0.612$, $p < 10^{-38}$; **Reviewer Fig.**
**9**). We believe this supports the validity of this approach for inferring view invariance. Thus, we hope the
reviewer agrees it is useful to include this data as an Extended Data Figure. We underscore that we
understand that this analysis is by no means a substitute for directly testing view invariance, and, as the
reviewer asked, we now tried to make this extremely clear in the text:

Thus, unlike in the macaque, increasing view invariance was not observed for predicted invariance in the tree shrew. As
has been previously shown in macaque (Freiwald and Tsao, 2010) and rats (Tafazoli et al, 2017), direct testing within
each area is needed to determine whether view invariance is a hallmark of the tree shrew visual pathway (**Fig. 6h**).

And also in the discussion

“Finally, DNN-predicted indices of view invariance were as high in tree shrew area V2 as in more anterior areas (**Extended Data Fig. 4**), with the caveat that this needs to be confirmed with direct measurement of view invariance in future studies”

“However, future work is needed to expand these findings with additional stimulus sets such as those that include ethologically relevant objects, view invariance, binocular disparity and multimodal stimuli. In fact, studies in rats using highly controlled visual morphed objects designed to match luminosity across transformations led to the discovery of properties of object processing in the rat visual area LL and TO including view invariance (Vermaercke et al., 2014; Tafazoli et al., 2017). Such comparisons in the tree shrew would further shed light into evolutionarily conserved properties for object vision that places this species functionally between the primates and rodents.”

[FIGURE REDACTED]

Reviewer Figure 9. DNN-predicted indices of view invariance are equally high across all tree shrew visual areas. (Replicated from Extended Data Fig. 5):

(a) [TEXT REDACTED] **(b)** Responses of three example cells from posterior V2 (left), posterior IT (middle) and anterior IT (right) to 50 objects (columns) each at 24 different views (rows). Top panel show actual responses, bottom panel shows responses predicted from an AlexNet model built from responses to 1593 images (see panel a). **(c)** Same as (b) but for predicted responses of example tree shrew neurons from all areas. **(d)** Histograms of invariance indices (*Methods*) of macaque V2, posterior and anterior IT neurons, calculated from actual responses (left) and predicted responses (right). Vertical lines indicate means. **(e)** Same as d, but for predicted responses across all tree shrew areas.

**IMPORTANT ISSUES**

 1) Page 3, lines 120-123. “Cells with ON and/or OFF receptive fields were clearly present in all areas except
 TP (Fig. 1f). Surprisingly, this included the two most anterior areas TI-ITi and ITr ...”. Fig. 1f is far from ideal
 to understand how the RF structure changes across the areas. I think it would be more informative to plot
 3 bars for each area: one with the fraction of RFs with ON regions, one with the fraction of RFs with OFF
 regions and one with the fraction of RFs with both ON and OFF regions.

However, it would be even better to carry out an analysis of the structure of the RFs to better quantify: 1)
 their complexity; and 2) their sharpness. One way to do so would be to measure the contrast of the RFs
 and to count the number of distinct lobes in the RFs as we did in (Matteucci et al., 2019; Matteucci and
 Zoccolan, 2020). Please refer to Fig. 5 in Matteucci et al 2019 for further details. Performing such
 quantification would allow to better support a statement like: “Surprisingly, this included the two most
 anterior areas TI-ITi and ITr; in contrast, corresponding areas in the anterior primate temporal lobe show
 largely spatially invariant responses”.

In fact, unless one considers the really highest stages of the monkey ventral stream (anterior IT), it is not
 so surprising for a middle-to-high level visual area to yield some sort of RF using reverse correlation. Even
 if the stimulus/response relationship is nonlinear (e.g., because of spatial invariance), visual neurons often
 retain a linear component of their responses and such linear residue may show up as a light or dark region
 when mapping RFs with sparse or dense noise. The more important question is whether these RFs are
 structured or not. In V1, simple cells with linear stimulus/response relationships will show a rich structure
 with multiple, crispy lobes (both ON and OFF). Higher order, more nonlinear units will display fainter RFs
 with a lower number of distinct lobes. This is what we found in the comparison between V1 and LL (one of
 the highest stages of the rat ventral stream) in Matteucci et al 2019. I believe it would be important to carry
 out this sort of quantification also here.

Following the reviewer’s suggestion, we have added a bar in **Fig. 1f** indicating the fraction of RFs with both
 ON and OFF regions (**Reviewer Fig. 10**).

**Reviewer Figure 10. On and OFF receptive fields were partially overlapping (Panel a replicated from Fig. 1f):**
 (a) Percentage of visually responsive units (cf. **Fig 1e**) exhibiting receptive fields (RFs) for each of the six recorded
 areas. Left (lighter, ON), center (darker, OFF) and right (ON/OFF) bars for each area. Dots represent results from
 individual recording sessions. (b) RF maps for example units, one per area. Rows are maps for ON RFs, OFF RFs and
 difference between ON and OFF RFs.

Exploring the structure of RFs is a very attractive idea. Unfortunately, we followed the Allen Institute
 approach of mapping receptive fields, which presents local sparse noise with alternate frames featuring
 white pixels on a gray field and black pixels on a gray field. Accordingly, our calculations of “ON” and “OFF”
 receptive fields are quite separate and based on the even and odd frames respectively. We found that for
 most cells that had a defined RF in both “ON” and “OFF” conditions, those RFs were very similar to each
 other. Fig. 9b shows six examples, one from each area. When we subtracted the “OFF” responses from the
 “ON” response (bottom row), no obvious lobe structure emerged for the majority of the cells we recorded
 from in tree shrews.

2) Page 4. Line 160-161 “V2 had the largest proportion of cells responding to the texture and/or noise
stimuli”. This quantification of explained variance is of course informative about the ability of cells in a
population to differentiate among textures but it does not provide a direct measure of how discriminable the
textures are based on the population activity. Also, I believe that Fig. 2i show the mean (or median?) of the
explained variance across the population, so it does not tell how well, collectively, the neuronal ensemble
is encoding the textures. So, if the goal is to support a statement like: “the observation that activity in V2
encoded texture family identity earlier than that in V1”, I would recommend carrying out a real decoding
analysis. As a reference, the authors could take a recent study of textures encoding in mouse visual cortex
(Bolaños et al., 2024). As done in this paper, they could use a d' index to measure the statistics. They could
also carry out a population decoding analysis using a binary classifier probing how well each population
can discriminate between texture and noise. Finally, they could run a population classification analysis to
check how well each pair of texture family is discriminated by the population activity in each area.
Incidentally, it is interesting to point out that, based on the results of Bolanos et al, 2024, in the mouse, it is
area LM to carry out more info about textures, as compared to V1. And LM has been equated to monkey
V2 by several anatomy studies (Gao et al., 2010; Wang et al., 2011, 2012). I believe this consistency among
monkeys, mice and tree shrews should be mentioned somewhere in the manuscript.
We thank the reviewer for this important suggestion. We have now performed a population decoding
analysis using a naive Bayes classifier to determine how well each population can discriminate between
texture and noise stimuli as well as gratings (**Reviewer Fig. 11**). For static grating stimuli we additionally
performed the decoding analysis for spatial frequency, orientation, and phase. We found that V2
outperforms all the other areas on texture stimuli (while it is on par with V1 for decoding grating identity).
We agree that this is an important analysis to compare our results with prior findings in monkey and rodents.
We now include this comparison to LM in rodents in our summary schematic (**Fig. 6h**). This schematic also
includes the provided references to LM (Gao et al., 2010; Wang et al., 2011, 2012; Bolaños et al., 2024).

**Reviewer Figure 11. Population decoding for gratings, texture and objects:**
Decoding performance for individual identity as a function of number of cells used by the classifier for static gratings, textures and object stimuli.

MINOR ISSUES

1) Page 3, Lines 125-130 "... receptive field positions were concentrated in a small portion of the screen ... Surprisingly, clustering of receptive fields was apparent in all areas studied, even TI-ITi and ITr ... maintain retinotopic organization; again, this contrasts with the primate, where retinotopy is absent in anterior temporal cortex." Two comments here:

A) The fact that "receptive field positions were concentrated in a small portion of the screen" and "clustering of receptive fields was apparent in all areas studied" does not mean anything per se, because the level of overlap between RF positions will depend on the amount of cortical surface spanned. And this, in turn, will depend on the inclination of the probe during the recording session. A probe inserted perpendicularly to the target area will not allow spanning much cortical surface and, because of the retinotopy, RFs will all tend to be close to each other. A probe inserted diagonally, with a large angle, will span instead a lot of cortical surface and therefore the RFs will span a large portion of the visual field. It is unclear here whether such concentration of RF centers on very similar locations is simply due to the penetration being very much orthogonal to the cortical surface. Please clarify.

Thank you, we see the reviewer's point that the receptive field locations depend strongly on the probe insertion angle in the cortex. For all our recording sites, the probe penetration was chosen to maximize recordings in each region. For that reason, the receptive fields tended to be close and correspond to a small portion of the screen as expected based on retinotopic cortical maps. For clarity, we have changed the statement referred to by the reviewer to state:

"receptive field positions were concentrated in a small portion of the screen, corresponding to our electrode penetration of the cortical surface..."

Furthermore, we have removed the following statement:

"again, this contrasts with the primate, where retinotopy is absent in anterior temporal cortex"

B) Regarding this statement "even TI-ITi and ITr, though located at the anterior end of the tree shrew ventral stream, maintain retinotopic organization; again, this contrasts with the primate, where retinotopy is absent in anterior temporal cortex", two comments:

First, it is too strong to say that there is no retinotopy in IT. While early studies in IT, as the one cited by the authors, reported very large RFs, spanning tens of deg, at least 3 studies have found very wide variations in RF size in IT, including RFs with just a few deg of size. Please refer to: (Op De Beeck and Vogels, 2000; DiCarlo and Maunsell, 2003; Zoccolan et al., 2007). It is true that there is a bias for RFs to be located close to the fovea but there is variation of several deg in their location (see Fig 6 in Op De Beeck and Vogels, 2000). So, this statement should be corrected to better reflect the literature and toned down.

Yet, it is true that, if you look at the medians across areas, as well as at the example RFs in fig. 1g, the increase from V1 to deeper areas is minimal. So, this is a place where some interesting comparison can be draw with what found along the rat ventral stream, where, instead, a very sharp increase of RF size was reported. Median RF size double from V1 to LL in (Tafazoli et al., 2017) and almost doubled from V1 to TO in (Vermaercke et al., 2014). Also along the mouse visual cortical hierarchy the mean RF size increases dramatically (Wang and Burkhalter, 2007; Siegle et al., 2019), although in the mouse it is hard to restrict this analysis to ventral regions only (given that, in this species, compared to the rat, the ventral stream is way less understood at the functional level). I think that this comparison also with rodent data should be mentioned, given that it may suggest a larger affinity between monkeys and rats/rodents than monkeys and screws, at least for some specific functional properties. I also think that it would be interesting to add these comparisons to Fig. 5g.

We understand the reviewer's point that we have oversimplified the work in the primate IT to suggest that retinotopy is completely absent. As stated, prior work has demonstrated a range of receptive field sizes depending on the recording location within the IT cortex. We now removed the sentence. Furthermore, we agree with the reviewer that the difference between mouse/rat receptive field sizes and tree shrew field sizes is interesting.

2) Analysis of orientation tuning (Fig. 2a-d). The trend found for the orientation tuning (larger in V1/V2 and smaller in deeper areas) is an interesting result, but I would recommend the authors to expand the interpretation of this finding through a deeper comparison with the monkey, rodent and modeling literature.

In fact, the question of whether orientation tuning should increase or decrease along a ventral hierarchy
does not have a so obvious answer. Some rodent studies have indeed assumed that a ventral stream,
given the need to process fine details of the image features, should retain or even amplify tuning for
orientation. See for instance (Marshel et al., 2011).

By contrast, we have shown in (Matteucci et al., 2019) and, later, in (Muratore et al., 2022), that, along an
object processing hierarchy (e.g., a deep net like VGG-16,) orientation tuning and, more in general,
information about orientation, does decrease in the deeper layers, after an initial increase across the very
initial few layers. As indicated by the authors, this is indeed consistent with what found along the monkey
ventral stream. But, rather than referring to the single study they have cited here, they could refer to the
meta-analysis of the monkey literature we carried out in Matteucci et al, 2019, where we pooled together
data about orientation tuning from about 15 studies in monkey V1, V4 and IT (see Table 1 and Fig. 7D).
Finally, in our study we did find a sharp decrease in orientation tuning along the rat ventral stream, which
is very consistent with the one found by the authors in the tree shrew. It would be interesting however, if
the authors, in addition to compute their metric based on explained variance, computed also the more
traditional **OSI index**. This would allow a more direct comparison with our meta-analysis of former monkey
studies and our finding on the rat. Also, it would perhaps be interesting this comparison to Fig. 5g.

We are glad that the reviewer finds this result interesting as do we. The meta-analysis referred to by the
reviewer is indeed a very useful reference for comparing to our finding in tree shrews about orientation
tuning across the hierarchy to other species, and we now cite this reference. We state:

These findings are roughly consistent with those found in the primate and rodent ventral stream, where orientation tuning
is especially prominent in early visual areas and then sharply decreases in later areas (David et al, 2006; Marshel et al.,
2011; Matteucci et al., 2019; Muratore et al., 2022).

We have expanded our summary schematic in **Fig. 6h** to include rats and to include this comparison of
orientation tuning. As suggested, we followed the methodology from Matteucci and Zoccolan, 2020 to
calculate the OSI on our data. We found a similar general trend with higher OSI values in V1 and V2
compared to other areas (**Reviewer Fig. 12**).

Reviewer Figure 12. Distributions of orientation selective index.

3) Page 5, line 222-223 “we summed the total explained variance for each area by each AlexNet layer
across cells”. This should be described better. The text says “we summed” but this would imply that the
metric increases as a function of the number of cells and, since a different number of cells was recorded in
each area, this would be a problem. I would have expected the authors to average the explained variance
across cells rather than simply sum or perhaps take the medians of the distributions of explained variance
across the recorded populations. Clarify please.

Our metric does not increase with the number of cells, because we take the summed explained variance
and divide by the summed explainable variance. The resulting ratio is independent of the number of cells.

We agree that our sentence was confusing and now modify it to say:

“To compare explanatory power of different AlexNet layers across brain areas, we calculated the sum across cells within
each area of the variance explained by the various AlexNet layers and normalized these sums by the sum across cells of
their explainable variance.”

4) Page 7, lines 339-340 “they were comparatively small ... relative to the large spatially-invariant receptive
fields of primate anterior IT cells (mean = 24.5 deg)”. As mentioned already, this number comes from a very
old paper and has not been confirmed by more recent investigations. The most systematic study of RF size
in IT reports a means of about 10 deg (Op De Beeck and Vogels, 2000). Other studies have shown RFs
that are even smaller - just a few degs. This statement should be toned down. It is true that the increase of
RF along the shrew ventral stream is surprisingly modest, compared to both monkey but also rats and mice.
Still, the monkey RF size in IT should not be overestimated.

We apologize for the oversimplification of the IT receptive field size as mentioned before. As the reviewer
states, this was to serve as a comparison for the relatively modest increase in receptive field sizes that we
observed in the tree shrew. We now modify the statement to state:

“While receptive field sizes were the largest in TI-ITi and ITr, they were comparatively small (mean = 5.6° and 6.1°
respectively) relative to the many large spatially-invariant receptive fields of primate anterior IT cells (spanning on average
10-20° (Ito et al, 1995, Op De Beeck and Vogels, 2000)”.

5) Fig. 5g. Given that object/shape processing has been studied way more systematically and carefully in
rats than in mice, I find it odd that the table include mice but not rats. A row describing the findings on rats
should be added. Under the column view invariance, the entry should report yes: LL and TO, based on
(Vermaercke et al., 2014; Tafazoli et al., 2017). Also, the entry of the texture column should be corrected
and report LM, based on (Bolaños et al., 2024).

Thank you for the great suggestion. We apologize for not including rats in our schematic. We
have now added an additional row for rat in Fig. 6h (**Reviewer Fig. 13**). We added the entry to
report the work showing view invariance in rat LL and TO. We modified the texture column to
report LM in mice and added this reference as well.

[FIGURE REDACTED]

Reviewer Figure 13. [TEXT REDACTED]

**Referees' comments:**

We sincerely thank the reviewers for their comments and suggestions throughout the entire review process.
We believe their feedback has meaningfully improved the paper, and we fully acknowledge that the final
version owes much to their careful and constructive input.

**Referee #1 (Remarks to the Author):**

The new article is well revised, and the addition of new data from macaque to provide quantitative
comparison is excellent. Figure 6g is a fundamental conclusion that will be often reprinted and examined.
The authors have done a good job addressing all the concerns.

I have only a very small comment. In Figure 6g, there are no black lines on the primate side (face cell
decoding). I don't think it is critical that they be added, but as a reader, I found myself not understanding
why they weren't there. Some small note in the caption as to why they are present on the tree shrew side
and not on the primate side would be very helpful. (If you can do the same analysis and add them that
might be best, because I think that panel is going to be reproduced in many talks, reviews, textbooks, etc.)
We are very grateful to the reviewer for their kind words and for pointing out the missing black line on the
primate side of Figure 6g. We have now added the black line for face decoding in the macaque data and
updated the figure caption accordingly. We agree that this addition helps clarify the figure and strengthens
the comparison between the two species, and we appreciate the helpful suggestion.

**Reviewer Figure 1. (Replicated from Fig. 6g):**
Decoding performance for individual object identity (dashed lines) or face identity (solid lines) as a function of number of cells used by the classifier for tree shrew (left) and macaque (right). Note the overlap of the two lines for TI-ITi. Black lines indicate decoding performance for face identity using only face cells (t-score greater than 5). Dashed gray lines: chance level for object decoding.